# Molecular and topological reorganizations in mitochondrial architecture interplay during Bax-mediated steps of apoptosis

**Nicholas R Ader[1,2], Patrick C Hoffmann[1], Iva Ganeva[1], Alicia C Borgeaud[1], Chunxin Wang[2], Richard J Youle[2], Wanda Kukulski[1]***

[1]Cell Biology Division, MRC Laboratory of Molecular Biology, Cambridge, United Kingdom; [2]Biochemistry Section, Surgical Neurology Branch, National Institute of Neurological Disorders and Stroke, National Institutes of Health, Bethesda, United States

**Abstract** During apoptosis, Bcl-2 proteins such as Bax and Bak mediate the release of pro-apoptotic proteins from the mitochondria by clustering on the outer mitochondrial membrane and thereby permeabilizing it. However, it remains unclear how outer membrane openings form. Here, we combined different correlative microscopy and electron cryo-tomography approaches to visualize the effects of Bax activity on mitochondria in human cells. Our data show that Bax clusters localize near outer membrane ruptures of highly variable size. Bax clusters contain structural elements suggesting a higher order organization of their components. Furthermore, unfolding of inner membrane cristae is coupled to changes in the supramolecular assembly of ATP synthases, particularly pronounced at membrane segments exposed to the cytosol by ruptures. Based on our results, we propose a comprehensive model in which molecular reorganizations of the inner membrane and sequestration of outer membrane components into Bax clusters interplay in the formation of outer membrane ruptures.

**Editorial note:** This article has been through an editorial process in which the authors decide how to respond to the issues raised during peer review. The Reviewing Editor's assessment is that all the issues have been addressed (see decision letter).
DOI: https://doi.org/10.7554/eLife.40712.001

**\*For correspondence:**
kukulski@mrc-lmb.cam.ac.uk

## Introduction

Controlled cell death mediated by the mitochondria is a critical check on inappropriate cell proliferation (*Labi and Erlacher, 2015*; *Youle and Strasser, 2008*). Pro-apoptotic members of the Bcl-2 protein family, including Bax, Bak, and the less studied Bok, are central to facilitating the necessary release of apoptotic factors, such as cytochrome *c* and Smac/DIABLO, from the mitochondria into the cytosol (*Jürgensmeier et al., 1998*; *Ke et al., 2018*). In healthy cells, Bax cycles between the surface of mitochondria and the cytosol, while Bak resides mostly on mitochondria (*Edlich et al., 2011*; *Griffiths et al., 1999*). Upon activation by apoptotic stimuli, Bax/Bak stably inserts into the outer membrane of the mitochondria. This step leads to permeabilization of the outer membrane, which is required for release of the apoptotic factors from the intermembrane space, the compartment formed from the intracristae and peripheral space (*Lovell et al., 2008*).

Bax has been long known to form membrane pores and ruptures *in vitro* (*Antonsson et al., 1997*; *Basañez et al., 1999*; *Schafer et al., 2009*; *Schlesinger et al., 1997*). In purified outer mitochondrial membranes, Bax-induced ruptures have been observed by electron cryo-microscopy (cryo-EM) (*Gillies et al., 2015*). Similar evidence for outer membrane ruptures in mitochondria of intact cultured cells has only been obtained recently. Super-resolution fluorescence microscopy (FM)

revealed that activated Bax forms rings devoid of outer mitochondrial membrane proteins, suggested to correspond to outer membrane ruptures several hundreds of nm in diameter (*Große et al., 2016*; *Salvador-Gallego et al., 2016*). The occurrence of such large ruptures has been confirmed by electron cryo-tomography (cryo-ET), and associated with the extrusion of mitochondrial DNA (mtDNA) through the opened outer membrane (*McArthur et al., 2018*).

While the mechanism of formation of these large ruptures remains elusive, the ability of Bax/Bak to associate into oligomeric assemblies appears to be essential for permeabilizing the outer membrane (*Antonsson et al., 2000*; *Nechushtan et al., 2001*; *Westphal et al., 2014*). The conformational changes that lead to Bax activation include insertion of a transmembrane helix into the outer membrane and subsequent dimerization of membrane-bound Bax (*Bleicken et al., 2014*; *Brouwer et al., 2014*; *Czabotar et al., 2013*; *Dewson et al., 2008*; *Dewson et al., 2012*). Further accumulation into larger Bax/Bak assemblies involves interactions via multiple, labile interfaces (*Uren et al., 2017*). In FM, the formation of these assemblies can be observed as small punctae on the mitochondria that coalesce into larger, mitochondria-associated cytosolic clusters that contain thousands of Bax/Bak molecules (*Nasu et al., 2016*; *Nechushtan et al., 2001*; *Zhou and Chang, 2008*). It is only poorly understood how the initial association of Bax/Bak molecules within the planar membrane rearranges into a three-dimensional cluster, and whether additional components are involved in forming the structures referred to as clusters (*Uren et al., 2017*). Further, the mechanism by which formation of these large clusters contributes to the release of apoptotic factors is not clear.

In addition to outer membrane rupturing, activation of Bax/Bak has been implicated in inner mitochondrial membrane rearrangements, suggested to be required for efficient discharge of apoptotic factors trapped in the intracristae space (*Ban-Ishihara et al., 2013*; *Cipolat et al., 2006*; *Frezza et al., 2006*; *Scorrano et al., 2002*). The relationship between changes in inner membrane morphology, the formation of large outer membrane ruptures, and cytosolic Bax/Bak clusters is unclear.

Here, we used a set of correlative microscopy approaches, including electron tomography (ET) of resin-embedded as well as vitreous cells, to visualize the cellular structures associated with signals of GFP-tagged Bax. We thereby investigated membrane rupturing, cluster formation and inner membrane remodeling at high resolution. Our data suggest that these Bax-mediated events interplay to facilitate the release of apoptotic factors.

## Results

### Bax clusters form regions of ribosome-exclusion in the cytosol

To mimic Bax-mediated apoptosis in HeLa cells, we took advantage of the previous observation that overexpression of Bax can induce cell death by apoptosis (*Han et al., 1996*; *Pastorino et al., 1998*). When cells expressed cytosolic GFP-Bax in the presence of the caspase inhibitor Q-VD-OPh, we observed on average 77 min later (SD 69 min, N = 86 cells) that GFP-Bax translocated to the mitochondria, which displayed fragmentation typical for apoptosis, as expected (*Karbowski et al., 2002*; *Figure 1A and B*). On average, 102 min (SD 57 min, N = 92 cells) after the initial recruitment into diffraction-limited punctae (*Figure 1B*), larger, irregular foci of Bax appeared (*Figure 1C*). Cells representing these two stages were similarly frequent 14–18 hr after GFP-Bax transfection. We confirmed by immunofluorescence that these stages coincided with the release of cytochrome *c* from the mitochondria (*Figure 1—figure supplement 1*). Of 42 cells expressing GFP-Bax, 9 contained diffraction-limited Bax punctae and displayed no or little cytosolic cytochrome *c* release (*Figure 1—figure supplement 1B*). Thirty-three cells contained larger GFP-Bax foci, of which 17 displayed partial and 16 complete cytochrome *c* release (*Figure 1—figure supplement 1C and D*, respectively). Consequently, for our further experiments, we chose 16 hr after GFP-Bax transfection as a time point that captures stages around cytochrome *c* release.

To visualize Bax clusters and associated mitochondrial membrane shape, we imaged resin-embedded cells by correlative FM and ET (*Ader and Kukulski, 2017*; *Kukulski et al., 2011*) (*Figure 2*). We targeted 82 GFP-Bax signals by ET and found that 79 of them localized adjacent to mitochondria (*Figure 2*; *crosses*). Further, of the 82 GFP-Bax signals imaged, 77 localized to dense regions in the cytosol that were devoid of other cytosolic features. In particular, they excluded the otherwise ubiquitously distributed ribosomes. These regions were irregular in shape and extended

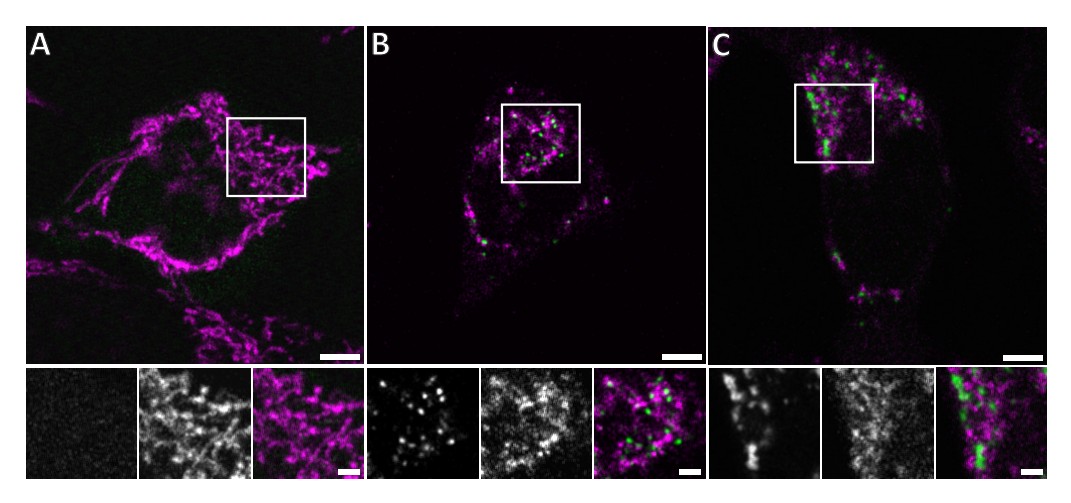

**Figure 1.** Live confocal fluorescence microscopy of HeLa cells overexpressing GFP-Bax. HeLa cells transfected with GFP-Bax (green) in the presence of Q-VD-OPh were imaged every 30 min for 24 hr after transfection. Cells were stained with MitoTracker Deep Red (magenta) prior to imaging. (**A**) Representative cell 9 hr after transfection. (**B**) Representative cell 14 hr after transfection, showing formation of GFP-Bax punctae. (**C**) Larger clusters of GFP-Bax in a representative cell 19 hr after transfection. White boxes indicate areas shown magnified below the large image. The three magnified images correspond to: GFP-Bax channel (left), MitoTracker Deep Red channel (middle), and merge (right). Scale bars: 5 µm (upper panel) and 2 µm (lower panel).

DOI: https://doi.org/10.7554/eLife.40712.002

The following figure supplement is available for figure 1:

**Figure supplement 1.** Immunofluorescence of cytochrome *c* release at different stages of GFP-Bax recruitment to mitochondria.

DOI: https://doi.org/10.7554/eLife.40712.003

over approximately 100 to 1300 nm. More intense GFP-Bax signals corresponded to larger ribosome-exclusion zones (*Figure 2F*). We thus conclude that these ribosome-exclusion zones in the cytosol comprise the Bax clusters previously observed by immuno-electron, scanning confocal, and super-resolution microscopy (*Große et al., 2016*; *Nasu et al., 2016*; *Nechushtan et al., 2001*; *Salvador-Gallego et al., 2016*; *Zhou and Chang, 2008*). We henceforth refer to these cellular structures as Bax clusters.

## Mitochondria near Bax clusters display outer membrane ruptures, influx of cytosolic content, and inner membrane restructuring

The mitochondria that we found near GFP-Bax clusters often exhibited substantial gaps in their outer membranes (*Figure 2*), which we henceforth refer to as ruptures. These ruptures were between 100 and 700 nm wide (mean 316 nm, SD 156 nm, N = 37). Of the 37 mitochondrial ruptures we found, 33 directly bordered the Bax clusters (*Figure 2*). Near the rupture, the remaining outer membrane appeared associated with the inner membrane at a similar distance as in non-ruptured regions. There were no outer membrane segments peeling off significantly from the inner membrane, or membrane segments loosely adhering to the remaining outer membrane. Although most ruptured mitochondria had single ruptures visible, occasionally two ruptures could be discerned at different regions of the same mitochondrion. Some of the mitochondria displayed large-scale concave indentations of their surfaces. Depending on their orientation within the tomographic volume, these indentations gave the false appearance of cytosolic content enclosed in a mitochondrion (*Figure 2F*, and *Figure 2—figure supplement 1*).

In 12 of the ruptured mitochondria, we observed ribosome-like structures, often several dozen, in the intermembrane space (*Figure 2F and J*; *red circles and white spheres*). In electron tomograms, ribosomes are easy to recognize because of their dense staining and ubiquitous presence in the cytosol (*Watson, 1958*). As mitochondrial ribosomes are confined to the mitochondrial matrix, we concluded that these were ribosomes that had leaked in from the cytosol through the outer membrane rupture. We also found 60 mitochondria near GFP-Bax clusters that had ribosomes in the

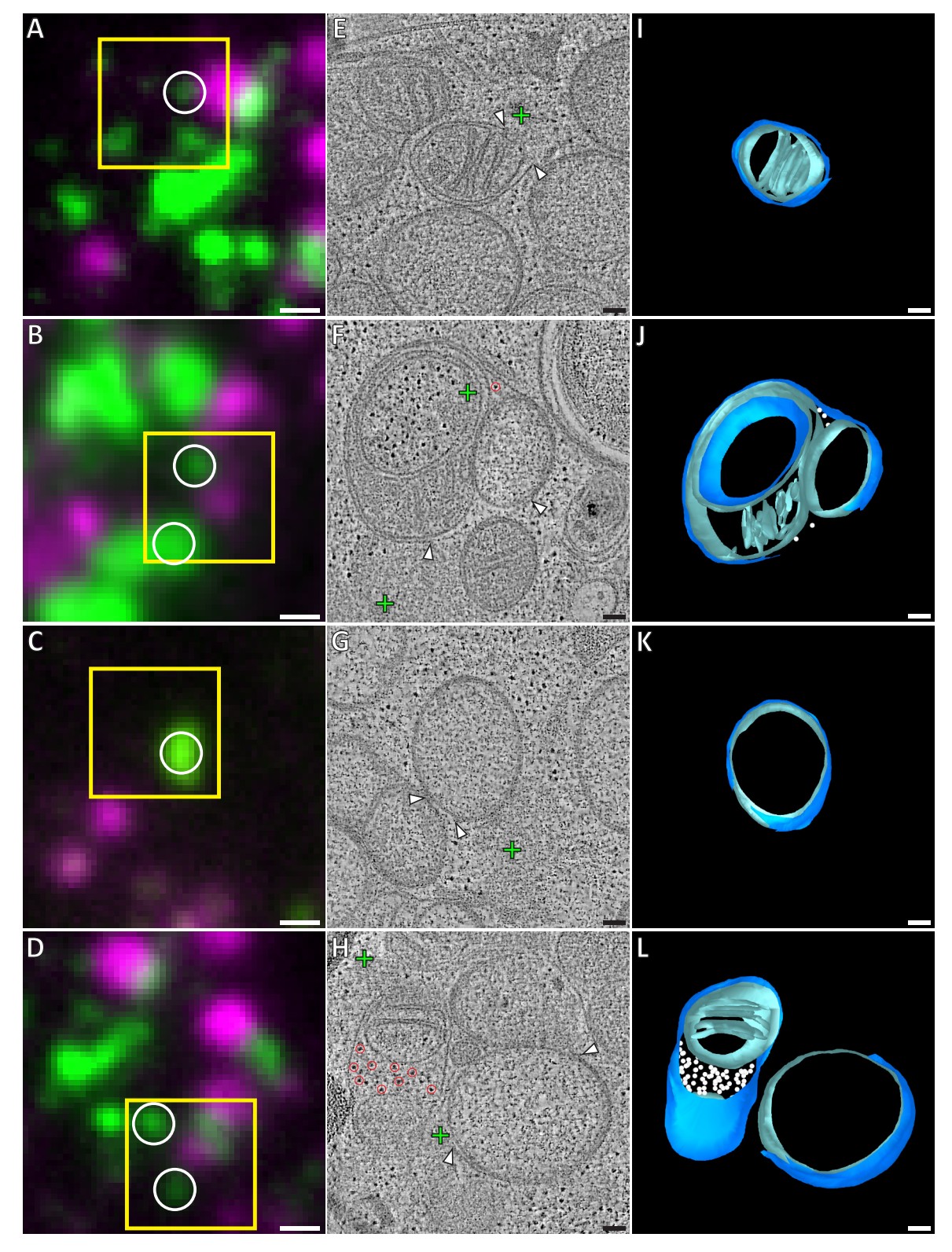

**Figure 2.** Correlative microscopy of resin-embedded HeLa cells overexpressing GFP-Bax. Gallery of GFP-Bax locations and the associated mitochondrial morphologies, 16 hr post-transfection with GFP-Bax, in the presence of Q-VD-OPh. (A–D) FM images of sections of resin-embedded cells. GFP-Bax (green) and MitoTracker Deep Red (magenta). Yellow squares indicate the field of view imaged by ET. White circles indicate GFP-Bax signals localized in electron tomograms. (E–H) Virtual slices from electron tomograms acquired at areas indicated by yellow squares in A-D,
*Figure 2 continued on next page*

 **eLIFE** Research Communication

Cell Biology | Structural Biology and Molecular Biophysics

*Figure 2 continued*

respectively. Red circles mark representative ribosomes in intermembrane space. White arrowheads indicate membrane ruptures. Green crosses indicate predicted positions of GFP-Bax signal centroids indicated by white circles in fluorescence micrographs. (I–L) 3D segmentation models of mitochondria in E-H, respectively. Outer membranes are in dark blue, inner membranes in light blue and ribosomes in the intermembrane space in white. Scale bars: 500 nm (**A–D**), 100 nm (**E–L**).

DOI: https://doi.org/10.7554/eLife.40712.004

The following figure supplements are available for figure 2:

**Figure supplement 1.** Representation of mitochondrial morphology in different views.

DOI: https://doi.org/10.7554/eLife.40712.005

**Figure supplement 2.** Drug-induced GFP-Bax recruitment to mitochondria in HCT116 cells causes outer membrane ruptures and inner membrane rearrangement similar to those induced in HeLa cells upon GFP-Bax overexpression.

DOI: https://doi.org/10.7554/eLife.40712.006

intermembrane space, but that had no outer membrane ruptures visible within the tomogram (*Figure 2H and L*; *red circles and white spheres*). The ribosomes in the intermembrane space suggested that many of the mitochondria we imaged had ruptures that were not contained within the imaged cell volume. Therefore, the presence of ribosomes in the intermembrane space offered indirect confirmation of outer membrane rupture, and indicated a relocation of cytosolic content into the intermembrane space upon outer membrane rupturing.

The ruptured mitochondria in our dataset showed a wide heterogeneity of inner membrane morphology. While some ruptured mitochondria displayed canonical cristae folding (*Figure 2E*), others lacked cristae over large areas of a smooth inner membrane (*Figure 2G and H*). Furthermore, we frequently observed more than one inner membrane compartment surrounded by a single outer membrane, indicating fragmentation of the inner membrane without concomitant outer membrane fission. In these cases, one matrix displayed canonical cristae shape, while the other matrix lacked cristae (*Figure 2F*). These observations indicate that, besides outer membrane ruptures, Bax activity induces fragmentation and restructuring of the inner membrane.

## Drug-induced apoptosis has similar effects on mitochondrial membranes to Bax overexpression

We next set out to test whether the mitochondrial restructurings we observed in HeLa cells upon overexpression of Bax were intrinsic hallmarks of apoptosis. We therefore analyzed Bax/Bak double knockout (DKO) HCT116 cells stably expressing GFP-Bax, in which we induced apoptosis with ABT-737, a BH3 mimetic pro-apoptotic compound (*van Delft et al., 2006*) (*Figure 2—figure supplement 2*). We found that the signals of GFP-Bax foci localized to ribosome-exclusion zones like in HeLa cells overexpressing GFP-Bax (*Figure 2—figure supplement 2G*). We also observed ruptured outer membranes, mostly (3 of 5 ruptures) near Bax clusters. The ruptures were, however, less frequent (5 ruptures for 45 GFP-Bax target signals) than in HeLa cells. These ruptured mitochondria displayed multiple matrices and unfolded inner membranes, similar to those in Bax-overexpressing HeLa cells (*Figure 2—figure supplement 2D–L*). Furthermore, 13 other mitochondria had multiple matrices, while no rupture was observed within the imaged cell volume. These results suggest that ribosome-excluding Bax clusters, ruptures in the outer membrane, as well as rearrangements of the inner membrane are characteristic of Bax activity independent of means inducing apoptosis.

## Bax clusters consist of a sponge-like meshwork

We next sought to obtain higher resolution details of Bax cluster organization by using cryo-ET. In tomograms of resin-embedded cells, the clusters appeared amorphous (*Figure 2*), but protein structures are best preserved in vitreous ice (*Dubochet et al., 1988*). We therefore used a correlative cryo-microscopy approach that allowed us to locate GFP-Bax clusters in vitreous sections of HeLa cells that were vitrified by high-pressure freezing (*Bharat et al., 2018*) (*Figure 3A,B,E and F*). In electron cryo-tomograms acquired at the predicted GFP-Bax locations, we found ribosome-exclusion zones in the cytosol, in agreement with our data from resin-embedded cells (*Figure 3C and G*) (N = 7 GFP-Bax signals). Within these exclusion zones, we could discern ultrastructural details that were not visible in the electron tomograms of resin-embedded cells (*Figure 3D and H*, and *Videos 1* and *2*). We found irregularly arranged plane and line segments that appeared to be part of a dense

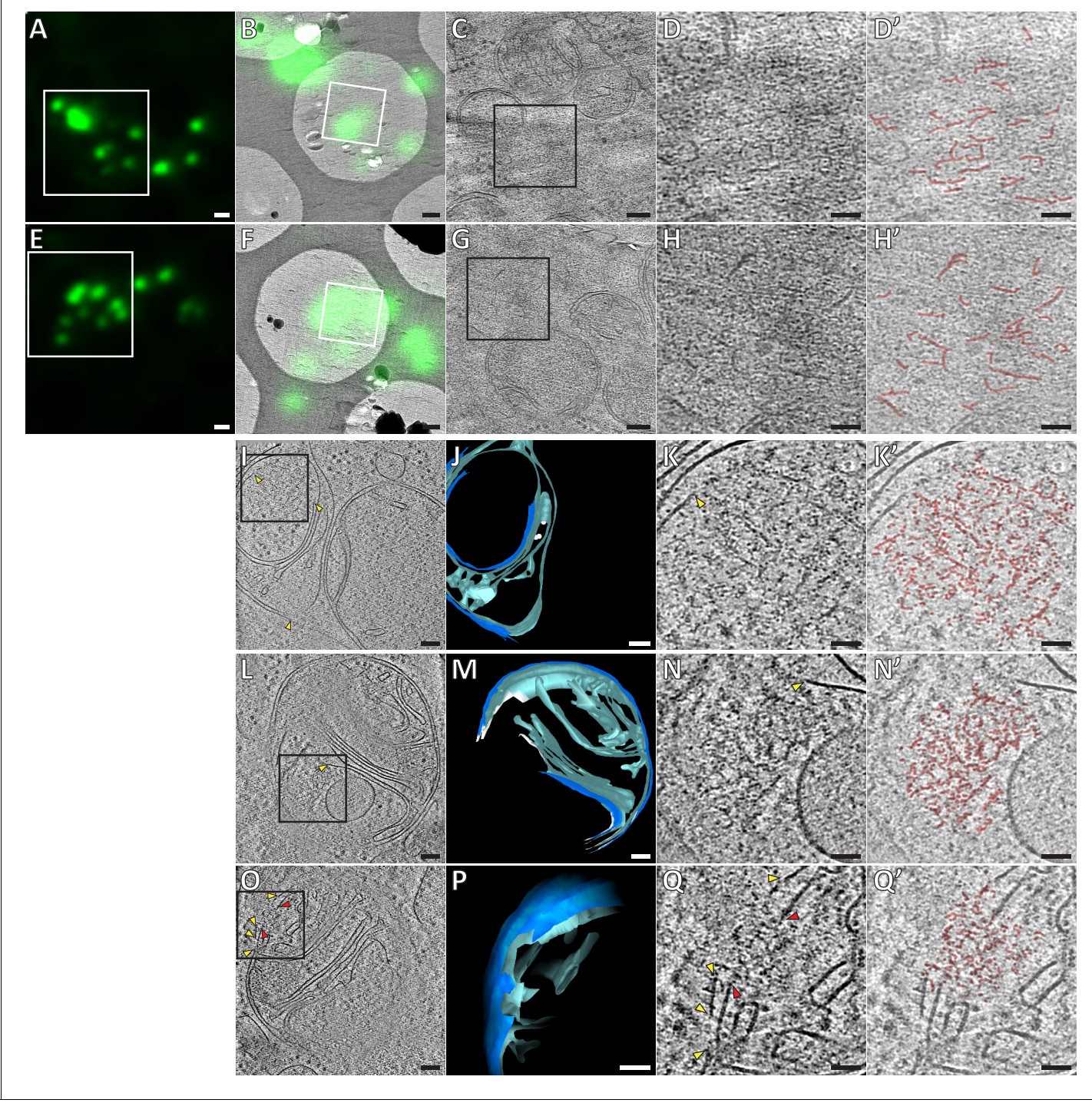

**Figure 3.** Ultrastructure of GFP-Bax clusters in HeLa cells visualized by correlative cryo-microscopy of vitreous sections, and by cryo-ET of FIB-milled cells. (A, E) Cryo-FM of vitreous sections of HeLa cells (high-pressure frozen 16 hr post-transfection with GFP-Bax). GFP-Bax signal in green. White squares indicate areas shown in B and F, respectively. (B, F) Cryo-EM overview images of areas shown in white squares in A and E, respectively. The corresponding cryo-FM images, transformed according to correlation procedure, are overlaid in green. White squares indicate areas imaged by cryo-ET. (C, G) Virtual slices through electron cryo-tomograms corresponding to white squares in B and F, respectively. Black squares indicate areas magnified in D and H, respectively. (D, H) Magnifications of virtual slices shown in C and G, respectively, areas corresponding to black squares. (D', H'): Annotation of images in D and H, respectively. Some of the structural features of the GFP-Bax cluster ultrastructure are highlighted in red. (I, L, O) Virtual slices through electron cryo-tomograms of HeLa cells (plunge-frozen 16 hr post-transfection with GFP-Bax), targeted by cryo-FM (see *Figure 3— figure supplement 1*) and thinned by cryo-FIB milling. Note that L and O show different virtual slices of the same mitochondrion rotated by 180°

*Figure 3 continued on next page*

*Figure 3 continued*

around the image y-axis. Black squares indicate areas magnified in K, N and Q, respectively. Yellow and red arrowheads indicate ruptured outer and inner membranes, respectively. (**J, M and P**) 3D segmentation model of mitochondria seen in I, L and O, respectively. Outer membranes are in dark blue, inner membranes in light blue and ribosomes in intermembrane space in white (**J**). White borders (**M**) indicate end of segmentation (see Materials and methods). Note that M and P show the same mitochondrion at different viewing angles and magnifications. (**K, N and Q**) Magnifications of virtual slices shown in I and L, respectively, areas corresponding to the black squares. Yellow and red arrowheads indicate ruptured outer and inner membranes, respectively. (**K', N' and Q'**) Annotation of images in K, N and Q, respectively. Some of the structural features of the cluster ultrastructure are highlighted in red. Scale bars: 1 μm (A, E), 500 nm (B, F), 100 nm (C, G, I, J, L, M, O, P), 50 nm (D, D', H, H', K, K', N, N', Q, Q').
DOI: https://doi.org/10.7554/eLife.40712.007

The following figure supplements are available for figure 3:

**Figure supplement 1.** Analysis of GFP-Bax clusters versus control regions in vitreous sections.
DOI: https://doi.org/10.7554/eLife.40712.008

**Figure supplement 2.** Cryo-FM targeted cryo-FIB milling of human apoptotic cells.
DOI: https://doi.org/10.7554/eLife.40712.009

network within the exclusion zones that corresponded to GFP-Bax localization (*Figure 3D' and H'*, and *Videos 1* and *2*; *red highlights*). We tested if the occurrence of these structural elements was specific to Bax clusters. For that, we compared areas that correlated to the presence of GFP-Bax signals to areas without GFP-Bax signal within the same tomogram, using an image analysis tool that detects ridge-like segments (see Materials and methods, and *Figure 3—figure supplement 1*) (*Steger, 1998*; *Wagner and Hiner, 2017*). We consistently found that the number of detected segments was higher in areas corresponding to GFP-Bax signals than in the areas that did not correlate to GFP-Bax signals (*Figure 3—figure supplement 1H*). These data suggest that Bax clusters are not amorphous, featureless structures, but that they contain elements indicative of a higher order ultrastructural organization.

While vitreous sections allow precise localization of fluorescent signals to electron cryo-tomograms (*Bharat et al., 2018*), artifacts induced by the sectioning process limit interpretability of structural details (*Al-Amoudi et al., 2005*). We therefore moved on to thinning cells grown on EM grids and vitrified by plunge-freezing using cryo-focused ion beam (FIB) milling (*Mahamid et al., 2016*; *Marko et al., 2007*). Prior to cryo-FIB milling, we screened these grids by cryo-FM to identify target cells that were transfected with GFP-Bax and were at the stage of Bax cluster formation. Furthermore, by targeting cell regions containing GFP-Bax clusters, we increased the likelihood that the clusters were contained in the thin lamellae produced by cryo-FIB milling (*Figure 3—figure supplement 2*). We then collected electron cryo-tomograms of mitochondria visibly identified in intermediate magnification maps of the lamellae (*Figure 3I and L*, *Figure 4A–C*). Adjacent to outer mitochondrial membrane ruptures in three different cells, we found six ribosome-exclusion zones that contained similar structural motifs as observed in vitreous sections. We therefore attributed these regions to correspond to Bax clusters (*Figure 3K,N and Q*, and *Videos 3* and *4*). These regions contained small planar segments, which manifest as lines in individual tomographic slices. The segments appeared irregularly connected to each other in a network (*Figure 3K', N' and Q'*, and *Videos 3* and *4*; *red highlights*). The average length of the segments was 21 nm (SD 5.6 nm, N = 59 segments from 3 clusters from two different cells). In

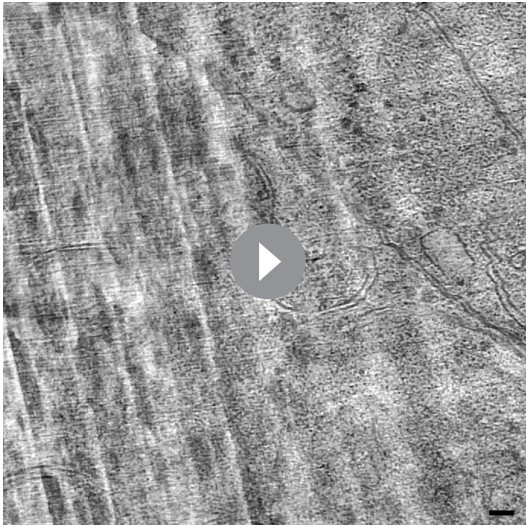

**Video 1.** Electron cryo-tomogram of GFP-Bax clusters obtained by correlative microscopy of vitreous sections, corresponding to *Figure 3C–D'*. Movie through virtual slices of electron cryo-tomogram. Movie pauses at the virtual slice shown in *Figure 3D and D'* to indicate structural features highlighted in red. Scale bar: 50 nm.
DOI: https://doi.org/10.7554/eLife.40712.010

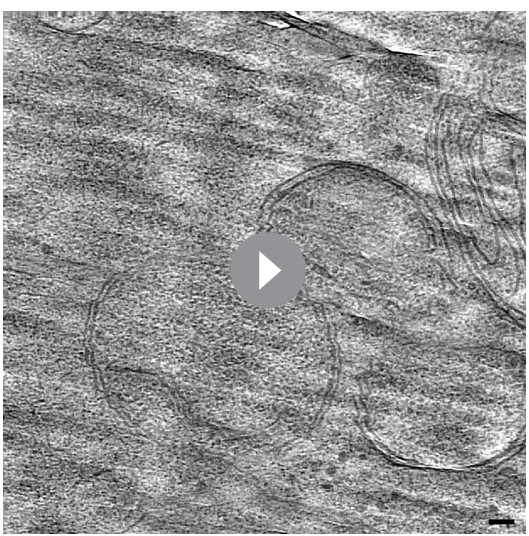

**Video 2.** Electron cryo-tomogram of GFP-Bax clusters obtained by correlative microscopy of vitreous sections, corresponding to *Figure 3G–H'*. Movie through virtual slices of electron cryo-tomogram. Movie pauses at the virtual slice shown in *Figure 3H and H'* to indicate structural features highlighted in red. Scale bar: 50 nm.
DOI: https://doi.org/10.7554/eLife.40712.011

addition, dot-like densities could be discerned at and between the segments (*Figure 3K', N' and Q'*, and *Videos 3* and *4*; *red highlights*). The network ultrastructure resembled a sponge with irregular fenestration, containing patches of high or low density.

Thus, using two different vitrification and two independent imaging methods, we identify structural motifs that suggest that Bax clusters contain higher order structures and their supramolecular organization resembles a sponge-like meshwork.

## Inner membrane flattening is most definite at outer membrane ruptures and inner membrane reshaping correlates with rupture size

We sought to use the superior preservation in cryo-ET to reveal details of the changes in membrane architecture occurring to apoptotic mitochondria. First, we inspected the ruptures in the outer mitochondrial membranes from five cells (*Figure 3I–Q* and *Figure 4*). The ruptured membrane bilayers displayed distinct, often sharp edges (*Figure 4D–I*) that were similar in thickness to the rest of the membrane. Some of the rupture edges appeared embedded into the cluster (*Figure 3K,N,Q* and *Figure 4F,H*). Furthermore, fragments of bilayer that appeared continuous with the outer membrane appeared also connected to the cluster (*Figure 3Q*; *yellow arrowheads*). In 11 of the 12 outer membrane ruptures we visualized by cryo-ET, the inner membrane appeared intact with no visible rupture. In only one case, we observed that both outer and inner membranes were ruptured, and a Bax cluster was protruding through the rupture into the mitochondrial matrix (*Figure 3O–Q'*). In the other 11 cases of ruptured outer membrane, substantial segments of the inner membrane were exposed to the cytosol at the site of the rupture (*Figure 4A–C*). In nine of these cases, there were no cristae protruding from the exposed inner membrane segment, and no intracristae spaces exposed to the outer membrane ruptures (*Figure 4A–C*). Thus, these segments appeared very smooth relative to the rest of the inner membrane, which displayed cristae of variable curvature that protruded into the matrix (*Figure 4J–L*).

We classified the ruptured mitochondria that we observed both by ET of resin-embedded cells and by cryo-ET of cryo-FIB milled HeLa cells, into three categories based on inner membrane morphology: Lamellar, approximately parallel cristae (N = 9), multiple matrices (N = 14), and mostly unfolded or short, tubular cristae (N = 20) (*Figure 4M*). It is possible that more mitochondria in our data set corresponded to the category with multiple matrices. The tomographic volumes are too thin to contain mitochondria in full and, therefore, we might not see all matrices. While we observed the smallest rupture sizes of approximately 100 nm in all three categories, increasingly larger ruptures were found for mitochondria with multiple matrices and with unfolded cristae, respectively (*Figure 4M*, p=0.0024). These results indicate that rupture size and the degree of inner membrane reshaping correlate with each other.

The mitochondria in the last category, which shared a similar degree of unfolded cristae and largely flattened inner membrane, appeared nearly spherical (*Figure 2G,H,K,L* and *Figure 4A–L*). We could thus estimate the total surface area of these mitochondria, and the surface area of the inner membrane that was exposed to the cytosol due to the rupture. The percentage of mitochondrial surface area that consisted of exposed inner membrane varied between 2% and 50% (mean total surface area: 1.15 $\mu m^2$, SD 0.41 $\mu m^2$, N = 19; mean surface area of exposed inner membrane: 0.21 $\mu m^2$, SD 0.19 $\mu m^2$, N = 19) (*Figure 4N*). Thus, rupture sizes varied largely at a given stage of inner membrane remodeling.

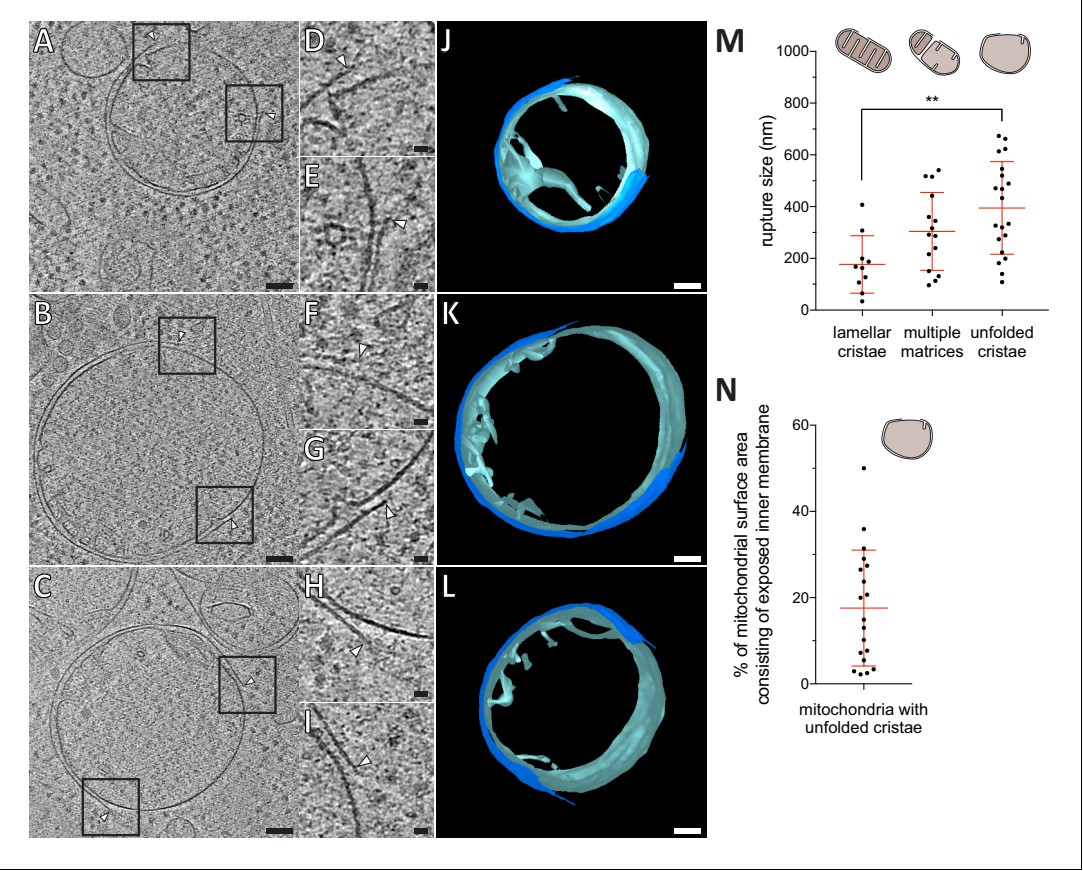

**Figure 4.** Mitochondrial outer membrane ruptures are accompanied by rearrangements of the inner membrane. (**A-C**) Virtual slices through electron cryo-tomograms of HeLa cells (16 hr post-transfection with GFP-Bax), thinned by cryo-FIB milling. Black squares indicate areas magnified in D-I, respectively. (**D–I**) Magnifications of the virtual slices shown in A-C, respectively, areas corresponding to black squares. White arrowheads indicate ruptured membranes. (**J–L**) 3D segmentation model of mitochondria seen in A-C, respectively. Outer membranes are in dark blue, inner membranes in light blue. (**M**) Quantification of rupture sizes, grouped according to inner membrane morphology category. Data points are from both ET of resin-embedded HeLa cells and from cryo-ET of cryo-FIB milled HeLa cells, all 16 hr post-transfection with GFP-Bax. Schematic representation of each category is shown above columns. Note that the 'lamellar cristae' category contains 10 ruptures from 9 mitochondria. p=0.0024 for lamellar cristae vs. unfolded cristae. The red lines indicate the mean and the standard deviation. For numerical data see *Figure 4—source data 1*. (**N**) The percentage of mitochondrial surface area consisting of exposed inner membrane, plotted for mitochondria with unfolded cristae (indicated by schematic in upper right corner). The red lines indicate the mean and the standard deviation. For numerical data see *Figure 4—source data 2*. Scale bars: 100 nm (**A–C** and **J–L**), 20 nm (**D–I**).

DOI: https://doi.org/10.7554/eLife.40712.012

The following source data is available for figure 4:

**Source data 1.** Numerical data presented in the graph shown in *Figure 4M*.
DOI: https://doi.org/10.7554/eLife.40712.013
**Source data 2.** Numerical data presented in the graph shown in *Figure 4N*.
DOI: https://doi.org/10.7554/eLife.40712.014

## The matrices of apoptotic mitochondria are dilute compared to non-apoptotic mitochondria

As cryo-ET relies on the inherent contrast of native macromolecules, differences in density within individual tomograms can be interpreted as differences in density of macromolecules. The mitochondrial matrix is a compartment of high protein concentration (*Kühlbrandt, 2015*). Therefore, mitochondrial matrices are expected to display a higher density in cryo-ET than the surrounding cytoplasm. However, in our data set the mitochondria with unfolded inner membranes did not appear different in density than the surrounding cytosol (*Figure 5A*). To assess this observation quantitatively, we measured the ratio of average matrix-gray value to cytosol-gray value in electron

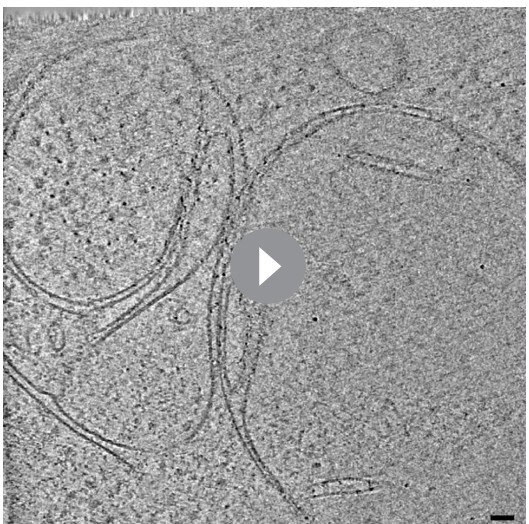

**Video 3.** Electron cryo-tomogram of GFP-Bax cluster obtained from cryo-FIB milled cells, corresponding to *Figure 3I–K'*. Movie through virtual slices of electron cryo-tomogram. 3D segmentation model of mitochondrion is shown as an overlay. Outer membranes in dark blue, inner membranes in light blue, and ribosomes in intermembrane space in white. Movie pauses at the virtual slice shown in *Figure 3K and K'* to indicate structural features highlighted in red. Scale bar: 50 nm.
DOI: https://doi.org/10.7554/eLife.40712.015

**Video 4.** Electron cryo-tomogram of GFP-Bax clusters and inner membrane rupture obtained from cryo-FIB milled cells, corresponding to *Figure 3L–Q'*. Movie through virtual slices of electron cryo-tomogram. 3D segmentation model of mitochondrion is shown as an overlay. Outer membranes are in dark blue, inner membranes in light blue. White borders indicate end of segmentation (see Materials and methods). Movie pauses at the virtual slice shown in *Figure 3N and N'*, and at the virtual slice shown in *Figure 3Q and Q'* to indicate structural features highlighted in red. Scale bar: 50 nm.
DOI: https://doi.org/10.7554/eLife.40712.016

cryo-tomograms of HeLa cells overexpressing GFP-Bax (*Figure 5A*) (N = 4 mitochondria). For comparison, we acquired electron cryo-tomograms of mitochondria in control HeLa cells that did not overexpress Bax and performed the same measurement (*Figure 5B*) (N = 5 mitochondria). The ratio was close to one in the cells overexpressing GFP-Bax, suggesting that the matrices of these mitochondria were similar in macromolecular density to the cytosol (*Figure 5C*). In mitochondria of control cells, the ratio was significantly lower (*Figure 5C*, p<0.0001), as expected for a compartment higher in macromolecular density than the cytosol. These results indicate that the mitochondria that had unfolded inner membranes in Bax-overexpressing cells had dilute matrices as compared to mitochondria in control cells.

## The organization of ATP synthases in apoptotic mitochondria exhibits localized changes

The dilute matrices allowed us to see individual protein complexes within the mitochondria of Bax-overexpressing HeLa cells, usually obscured by the high protein density (*Kühlbrandt, 2015*). In particular, ATP synthase heads were recognizable. As described by cryo-ET of purified mitochondria, densities characteristic for ATP synthases are localized at the ridges of cristae, where their distinct dimerization is thought to contribute to cristae structure (*Anselmi et al., 2018*; *Davies et al., 2012*; *Dudkina et al., 2010*; *Strauss et al., 2008*). We investigated the distribution of ATP synthases in apoptotic mitochondria (*Figure 5D,E*). ATP synthases were abundant on cristae (*Figure 5F and F'*; *matching arrowheads*). Albeit more rarely, ATP synthases were also present on shallow indentations of the boundary membrane, the region of the inner membrane directly opposed to the outer membrane (*Figure 5G and G'*; *arrowheads*). No ATP synthase heads were observed on the smooth regions of the inner membrane exposed to the cytosol by the ruptured outer membrane (*Figure 5H and H'*). Thus, the frequency of observing ATP synthases appeared to correlate with membrane curvature and the localization seemed to require an intact, adjacent outer membrane.

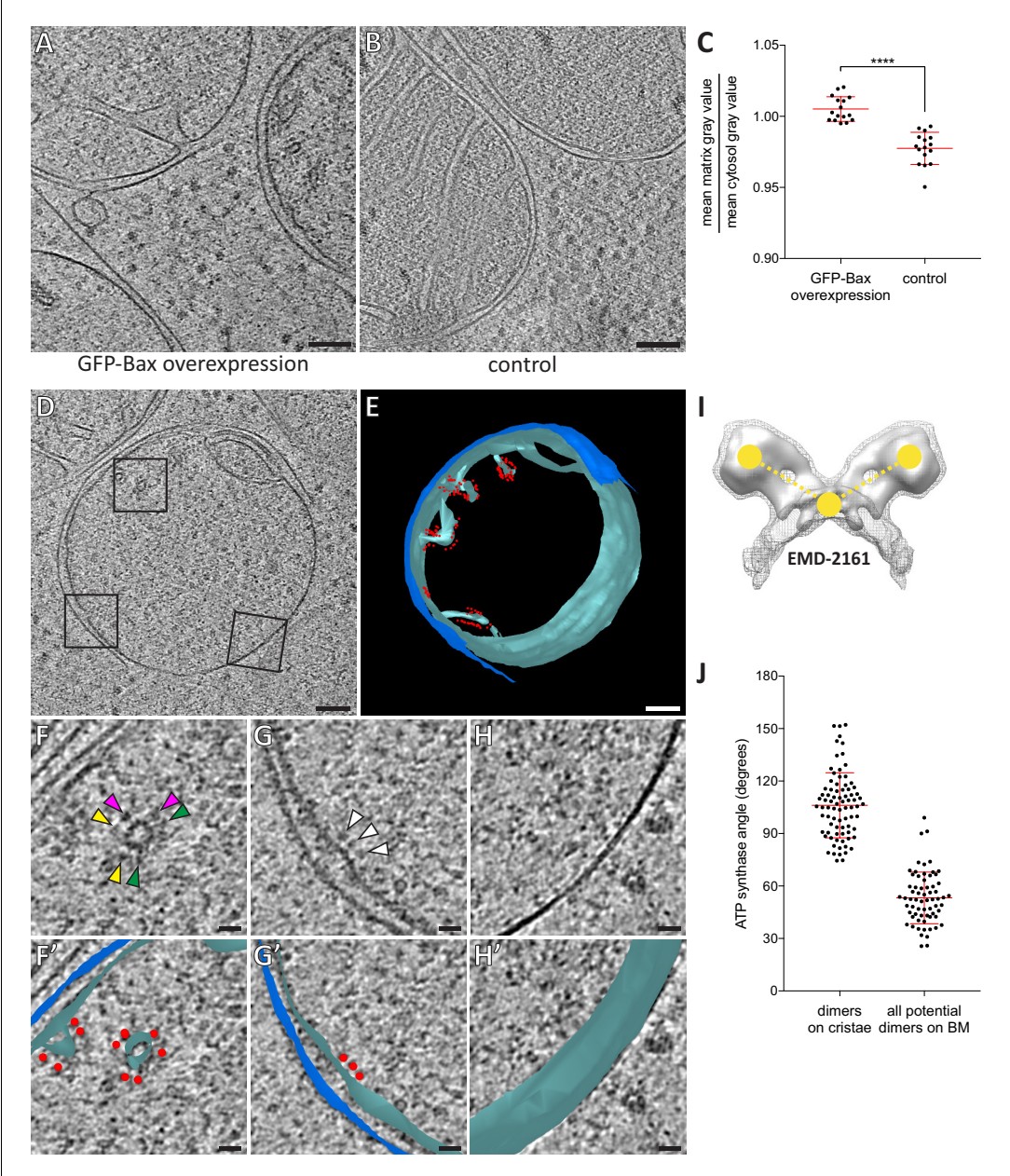

**Figure 5.** Dilution of the mitochondrial matrix and organization of ATP synthases visualized by cryo-ET of cryo-FIB milled HeLa cells. (A) Virtual slice through an electron cryo-tomogram of a HeLa cell (16 hr post-transfection with GFP-Bax) thinned by cryo-FIB milling. (B) Virtual slice through an electron cryo-tomogram of a control HeLa cell thinned by cryo-FIB milling, showing typical mitochondria in absence of GFP-Bax expression. (C) Quantitative analysis of the ratio between average pixel gray values in the matrix and average pixel gray values in the cytosol. A low value is attributed to a darker pixel, and a high value is attributed to a lighter pixel. p<0.0001 for comparison between mitochondria in GFP-Bax overexpressing and control HeLa cells. The red lines indicate the mean and the standard deviation. For numerical data see *Figure 5—source data 1*. (D) Virtual slice through an electron cryo-tomogram of a HeLa cell (16 hr post-transfection with GFP-Bax), thinned by cryo-FIB milling. Note that this is a different virtual slice of the tomogram shown in *Figure 4C*. Black squares indicate areas magnified in F-H. (E) 3D segmentation model of mitochondrion seen in D. Outer membranes are in dark blue, inner membranes in light blue and ATP synthase heads in red. (F–H) Magnified areas of the virtual slice shown in D, corresponding to the black squares. White arrowheads indicate ATP synthase heads. Arrowheads of matching color in F denote dimers of ATP synthases. (F'–H') Images from F-H shown with the segmentation model from E. Outer membranes are in dark blue, inner membranes in light blue, and ATP synthase heads in red. (I) Structure of the yeast ATP synthase dimer (EMD-2161, *Davies et al., 2012*), to illustrate how we measured the angle enclosed by ATP synthases heads and membrane (yellow points and dashed lines) for our analysis. (J) ATP synthase angles measured in dimers in cristae membranes, and between neighboring ATP synthases in the boundary membrane (BM). The red lines indicate the mean and the standard deviation. For numerical data see *Figure 5—source data 2*. Scale bars: 100 nm (A, B, D, E), 20 nm (F, F', G, G', H, H').

*Figure 5 continued on next page*

*Figure 5 continued*

DOI: https://doi.org/10.7554/eLife.40712.017

The following source data is available for figure 5:

**Source data 1.** Numerical data presented in the graph shown in *Figure 5C*.

DOI: https://doi.org/10.7554/eLife.40712.018

**Source data 2.** Numerical data presented in the graph shown in *Figure 5J*.

DOI: https://doi.org/10.7554/eLife.40712.019

We next investigated the dimeric states of the ATP synthases on cristae and boundary membranes. The ATP synthase dimer is reported to comprise an angle of 70–100° between the major stalks (*Davies et al., 2011*; *Hahn et al., 2016*). On cristae, dimers were readily discernable (*Figure 5F*). Within these dimers, we measured the angle enclosed by the two heads and the membrane between the two monomers (*Figure 5I*). The average angle was 106° (SD 19°, N = 81 dimers) (*Figure 5J*). Note that 106° measured in this way correspond to approximately 70° between the major stalks. On the boundary membrane it was not possible to unambiguously identify dimers among ATP synthases (*Figure 5G*). We therefore measured all possible angles between neighboring ATP synthases. These measurements thus included potential dimers as well as monomers positioned near to each other. For 52 ATP synthases on boundary membranes, we measured 66 angles between ATP synthase pairs (*Figure 5J*). The average angle was 53° (SD 15°), and only three ATP synthase pairs enclosed angles within the range we had measured for dimers in cristae, indicating that the majority of ATP synthases on the boundary membrane were not arranged into dimers similar to those on cristae. These results suggest that Bax-mediated flattening of the inner membrane is coupled to changes in the supramolecular organization of ATP synthases. These changes involve the dissociation of dimers into monomers upon unfolding of cristae, and clearance of ATP synthase heads from areas of smooth, cytosol-exposed inner membrane segments.

## Bax activity can result in mitochondrial matrices entirely devoid of outer membrane

Recently, leakage of mtDNA into the cytosol of apoptotic cells was reported to result from expulsion of inner membrane compartments through ruptured outer membranes (*McArthur et al., 2018*). In our correlative microscopy data from resin-embedded apoptotic Bax/Bak DKO HCT116 cells stably expressing GFP-Bax, MitoTracker signals localized to clumps of electron-dense compartments that were adjacent to GFP-Bax spots (*Figure 6A–D*). In cryo-ET data of these cells prepared by cryo-FIB milling, we also found single-membrane bound compartments that contained granular structures similar to those in the mitochondrial matrix (*Figure 6E* and *Wolf et al., 2017*), and highly curved membranes lined with particles reminiscent of ATP synthases (*Figure 6E–I*). To determine whether these compartments consisted of inner mitochondrial membranes, we tested whether the particles could correspond to ATP synthases by comparing them to the ATP synthases we identified in HeLa cells (*Figure 5*). We therefore measured the shortest distance from the center of the head to the membrane. The average distance was similar in both data sets (*Figure 6J*; HCT116: 12.02 nm, SD 1.69 nm, N = 65; HeLa: 12.07 nm, SD 1.00 nm, N = 65), and matched estimates from known ATP synthase structures (*Hahn et al., 2016*; *Hahn et al., 2018*; *Srivastava et al., 2018*). We concluded that these particles were likely ATP synthases, and hence these compartments corresponded to mitochondrial inner membranes lacking an outer membrane. These results indicate that Bax activity can result in complete removal of the outer membrane.

## Discussion

Three major ultrastructural processes have been associated with Bax/Bak activity and thus with the release of apoptotic factors from the intermembrane space of mitochondria during apoptosis. One is the necessity of Bax/Bak to oligomerize and form cytosolic assemblies known as clusters, which follows insertion of activated Bax/Bak in the outer mitochondrial membrane (*Große et al., 2016*; *Nechushtan et al., 2001*; *Uren et al., 2017*; *Zhou and Chang, 2008*). The second is the occurrence of large 'macropores' in the outer mitochondrial membrane, presumably for egress of apoptotic

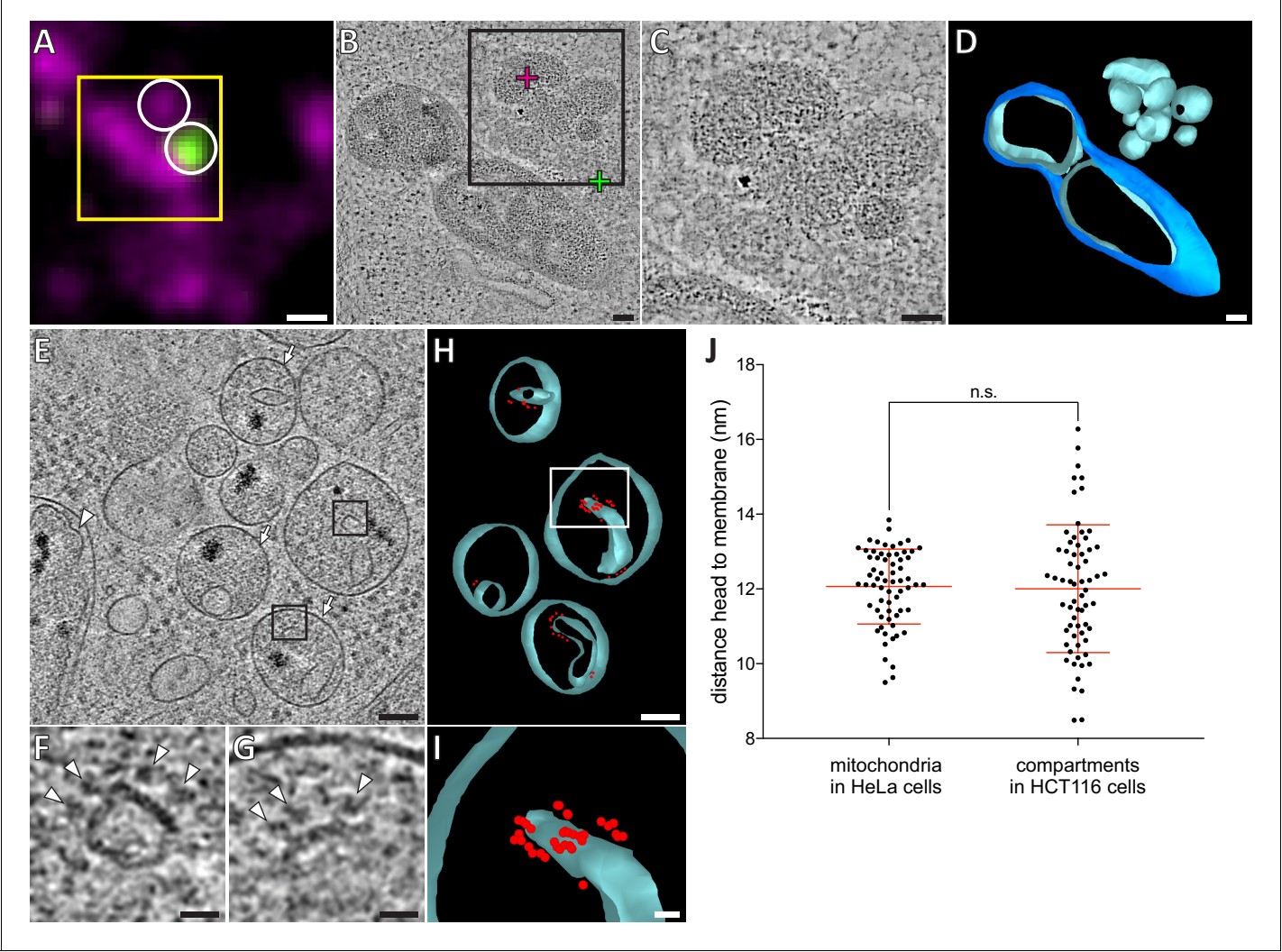

**Figure 6.** HCT116 cells treated with the apoptotic drug ABT-737 contain inner membrane compartments that are lacking the enclosing outer membranes. (A) FM of a section of resin-embedded Bax/Bak DKO HCT116 cells stably expressing GFP-Bax, treated with ABT-737 for 3 hr. GFP-Bax (green), MitoTracker Deep Red (magenta). Yellow square indicates the field of view imaged by ET, white circles indicate fluorescent signals of interest localized in electron tomograms. (B) Virtual slice through an electron tomogram acquired at area indicated by the yellow square in A. Green and magenta crosses indicate predicted position of GFP-Bax and MitoTracker Deep Red signal centroids, respectively, indicated by white circles in fluorescence micrographs. Black square indicates area magnified in C. (C) Magnified area of the virtual slice shown in B, corresponding to the black square. The image shows an accumulation of single membrane compartments near the GFP-Bax clusters. (D) 3D segmentation model of mitochondria and single-membrane compartments seen in B. Outer membranes are in dark blue, inner membranes in light blue. (E) Virtual slice through an electron cryo-tomogram of a cryo-FIB milled Bax/Bak DKO HCT116 cell stably expressing GFP-Bax treated with ABT-737 for 3 hr. Arrows indicate compartments reminiscent of mitochondrial inner membranes that appear to have no outer membrane.Arrowhead indicates an inner membrane within an intact mitochondrion. (F, G) Magnified areas of the virtual slice shown in E, corresponding to the black squares. White arrowheads indicate putative ATP synthase heads. (H) 3D segmentation model of compartments seen in E. Membranes are in light blue, putative ATP synthase heads in red. White box indicates magnified area in I. (I) Magnified area from white box in H, depicting the arrangement of putative ATP synthase heads. (J) Measured distances between head and membrane. Comparison between ATP synthases identified in mitochondria in HeLa cells, and putative ATP synthases in the compartments without outer membrane in HCT116 cells. The red lines indicate the mean and the standard deviation. For numerical data see *Figure 6—source data 1*. Scale bars: 500 nm (A), 100 nm (B–E, H), 20 nm (F, G, I).

DOI: https://doi.org/10.7554/eLife.40712.020

The following source data is available for figure 6:

**Source data 1.** Numerical data presented in the graph shown in *Figure 6J*.

DOI: https://doi.org/10.7554/eLife.40712.021

factors as well as mtDNA (*Große et al., 2016*; *McArthur et al., 2018*; *Riley et al., 2018*; *Salvador-Gallego et al., 2016*). The third is remodeling of the inner membrane, suggested to ensure complete release of the cytochrome *c* pool that resides predominantly in cristae (*Ban-Ishihara et al., 2013*; *Frezza et al., 2006*; *Scorrano et al., 2002*). There is, however, no unifying model on how these three major events are coupled to each other, to what extent each of them contributes to the release of apoptotic factors, and how Bax/Bak mediates all these events.

Here, we analyzed at high resolution the structural changes occurring at, and in, mitochondria of cultured human cells upon apoptotic Bax activity. We found that Bax clusters localized adjacent to ruptures of the outer membrane. We observed a single occurrence of the inner membrane being ruptured as well. Release of mtDNA has been recently associated with Bax/Bak activity and shown to involve inner membrane permeabilization (*McArthur et al., 2018*; *Riley et al., 2018*). Our observation provides visual evidence that Bax activity can rupture the inner membrane similarly to the outer membrane, albeit in our experimental setup this was a very rare event.

In HeLa cells apoptotic due to Bax overexpression, outer membrane ruptures varied between 30 and 700 nm in diameter, consistent with the sizes of Bax-rings and arcs reported by super-resolution FM (*Große et al., 2016*; *Salvador-Gallego et al., 2016*). This wide range of sizes could represent either different stages of progressive rupture widening (*Riley et al., 2018*), or inherent diversity at end stages of rupture formation. During drug-induced apoptosis in HCT116 cells, we also observed mitochondrial inner membrane compartments free from an encapsulating outer membrane, reminiscent of the recently reported, herniated inner membranes attributed a role in mtDNA signaling (*McArthur et al., 2018*; *Riley et al., 2018*). In HeLa cells, the mitochondrial surface area consisting of exposed inner membrane was highly variable but did not exceed 50%. The naked inner membranes we observed in HCT116 cells thus suggest that, in addition to inherent variability of rupture sizes, the degree of inner membrane exposure varies even more among cell types and/or means of apoptosis induction. Furthermore, our experiments were performed in the presence of caspase inhibitors to prevent detachment of cells from the substrate. This strategy allowed us to visualize events that might otherwise be very transient, but it may also affect timing or extent of some of the events observed.

Our data depicts ultrastructural and molecular details of the inner membrane architecture upon Bax activity. We observed fragmentation of the inner membrane compartments without outer membrane fission. In these instances, the intermembrane spaces were often enlarged and contained ribosomes, indicating influx of cytosolic content into the intermembrane space. Such a mixing of compartment content could potentially play a role in downstream apoptotic events that require interaction between components from both compartments, such as apoptosome formation (*Kim et al., 2005*; *Zhou et al., 2015*). In many cases, we observed that cristae locally unfolded into short, tubular protrusions and shallow ridges. These inner membrane remodeling events could be the result of changes in processing and/or activity of OPA1, leading to cristae disassembly and inner membrane fission (*Anand et al., 2014*).

We found that cristae remodeling was accompanied by disassembly of ATP synthases from dimers into loosely associated monomers. Furthermore, most inner membrane segments exposed to the cytosol by outer membrane ruptures appeared very smooth, displayed a consistently low curvature, and were devoid of ATP synthase heads. This shows that cristae unfolding and ATP synthase disintegration are maximal at outer membrane ruptures. We also observed a decreased density of macromolecules in the matrix of these mitochondria, indicating dilution of the matrix content. Matrix dilution could be caused by swelling and dilation of the inner membrane compartment, likely to generate turgor pressure and high membrane tension, which could facilitate cristae unfolding and disruption of the ATP synthase organization. Matrix dilution could also arise from efflux of matrix components in addition, or alternatively, to the volume increase.

The angular arrangement of ATP synthase dimers in cristae of Bax-affected mitochondria was similar to what has been reported for other species (*Davies et al., 2011*; *Hahn et al., 2016*), although our data presented a large range of dimer angles (*Figure 5J*). This range could be either due to inherent variability of ATP synthase dimers in human cells, or could reflect initial stages of dimer disassembly. It is thought that ATP synthase dimers induce membrane curvature, thereby contributing to the shape of cristae and to the proton-motive force (*Anselmi et al., 2018*; *Davies et al., 2012*; *Hahn et al., 2016*; *Strauss et al., 2008*). Disassembly of ATP synthase organization has been associated with loss of mitochondrial function and with aging (*Daum et al., 2013*). Here we show that a

local, distinctive two-stage disassembly of ATP synthases is part of Bax-mediated loss of cristae structure implicated in the release of apoptotic factors. This is particularly relevant as the loss of membrane curvature could help setting cytochrome *c* free, which is bound to cardiolipin in the intra-cristal space (*Scorrano et al., 2002*; *Speck et al., 1983*; *Vik et al., 1981*).

It is also worth noting that the smooth inner membrane exposed to the cytosol is remarkably similar to cryo-ET images from mouse embryonic fibroblasts shown by *McArthur et al., 2018*. This corroborates that the localized changes to the inner membrane we report here are general principles of Bax-mediated apoptosis.

Our quantitative analysis shows that the largest ruptures are found on mitochondria with almost completely unfolded cristae. This suggests a mechanism by which the inner membrane rearrangements could contribute to rupture formation: As the inner membrane flattens, the mismatch between inner and outer membrane surface area exerts pressure onto the outer membrane. This pressure could cause rupturing of outer membrane areas that are locally destabilized, for instance through accumulation of membrane-inserted Bax (*Westphal et al., 2014*). Further inner membrane flattening could widen initial ruptures. This mechanism could in principle generate large ruptures without removal of lipids from the outer membrane.

In this model, the rupture size would depend on the number and surface area of cristae to be unfolded in a given mitochondrion. We indeed observed that rupture sizes varied largely even at a given stage of inner membrane reorganization. This variability in rupture sizes could also be caused by additional factors potentially impacting rupture size. One such factor could be the amount of membrane-inserted, accumulated Bax molecules generating tension in the outer membrane (*Westphal et al., 2014*).

By correlative microscopy, we identified GFP-Bax signals to correspond to dense regions of the cytoplasm that we refer to as Bax clusters. While these data suggest that Bax is uniformly distributed within these clusters, it is possible that they contain additional components. These could be molecules originating from the cytosol, the intermembrane space or the outer membrane. We found that Bax clusters have a higher order organization consisting of interconnected planes or discs arranged in an irregular manner, reminiscent of a sponge-like meshwork. The lipid bilayer edges of the ruptures often appeared embedded in this meshwork or connected to its structural features. Some of the sharp edges within the meshwork resemble side views of membranes, suggesting that the clusters might contain patches of membrane. Previous models proposed that Bax/Bak cluster activity involves generating membrane tension, which is released by remodeling the bilayer of the planar outer membrane into a non-lamellar lipid arrangement (*Nasu et al., 2016*; *Uren et al., 2017*).

This remodeling could be aided by membrane sculpting proteins such as N-BAR domain proteins (*Gallop et al., 2006*). The N-BAR protein endophilin B1 interacts with Bax during apoptosis in cultured cells (*Takahashi et al., 2005*) and, also through interaction with Bax, causes vesiculation of liposomes in vitro (*Etxebarria et al., 2009*; *Rostovtseva et al., 2009*). Notably, the related N-BAR protein endophilin A1 can generate interconnected tubular membrane networks (*Ayton et al., 2009*; *Simunovic et al., 2013*). Furthermore, lipids such as cardiolipin and ceramides were attributed roles in supporting Bax activity (*Jain et al., 2017*; *Kuwana et al., 2002*).

Thus, we speculate that the higher order meshwork we observe for Bax clusters is a result of Bax reshaping outer membrane patches from a lamellar topology into a non-lamellar bilayer network, similar to sponge-like lipid cubic phases (*Valldeperas et al., 2016*). This model would suggest that ruptures might be formed through removal of lipids from the outer membrane. It would additionally explain how the clusters form: Bax oligomerization requires the interaction with membranes (*Bleicken et al., 2010*), while clusters occupy a volume in the cytosol. Therefore, there must be a transition from accumulation of Bax molecules in the membrane plane to a three-dimensional cluster consisting of Bax molecules and potentially other components. Following association within the outer membrane, Bax might progressively deform the membrane into a meshwork-like structure, which grows as more Bax molecules accumulate and serves as a sink for outer membrane components (*Uren et al., 2017*). Thus, the formation of the wide range of rupture sizes observed by others and us might be a consequence of two mechanisms: Cristae unfolding leading to flattening of the inner membrane, and sequestration of outer membrane components into Bax clusters (*Figure 7*).

In summary, we reveal molecular and morphological details of the effects of Bax activity on inner and outer mitochondrial membranes, suggesting how they collectively contribute to the release of

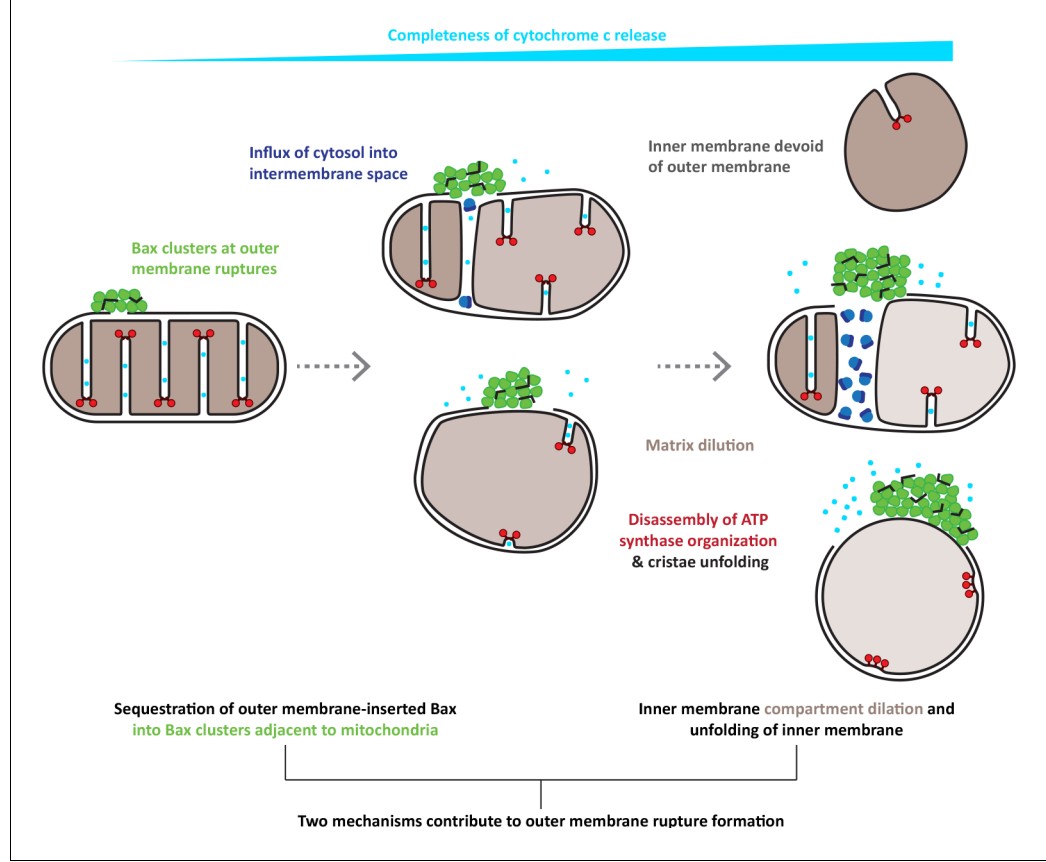

**Figure 7.** Model for the interplay of inner and outer membrane reorganization during Bax-mediated steps of apoptosis. Bax clusters form at outer membrane ruptures. Clusters display a higher order organization of their components. Ruptures allow influx of ribosomes and thus mixing of cytosolic and intermembrane content. As rupture size increases, the inner membrane remodels through fragmentation and cristae unfolding. Dilution of the mitochondrial matrix likely supports dilation of the inner membrane compartment. Inner membrane reshaping is accompanied by disassembly of ATP synthase dimers into monomers and a complete clearance of ATP synthases from regions of inner membrane that are exposed to the cytosol, and that are maximally flattened. The degree of inner membrane exposure varies, and is maximal in HCT116 cells, where inner membrane compartments devoid of any outer membrane can be found.

DOI: https://doi.org/10.7554/eLife.40712.022

apoptotic factors from mitochondria. Our study provides a comprehensive model on how reorganizations of the supramolecular architecture of membranes interplay to drive apoptosis (*Figure 7*).

## Materials and methods

**Key resources table**

| Reagent type (species) or resource | Designation | Source or reference | Identifiers | Additional information |
|---|---|---|---|---|
| Cell line (*Homo sapiens*) | Bax/Bak DKO HCT116 GFP-Bax | this paper | | Cell line generated by transfection of GFP-Bax and selection for stable expression in Bax/Bak DKO HCT116 line generated in PMID:22056880 |
| Cell line (*Homo sapiens*) | HeLa/wtOTC, TetOn promoter | PMID:24149988 | | Cell line maintained in Richard Youle lab. |

*Continued on next page*

*Continued*

| Reagent type (species) or resource | Designation | Source or reference | Identifiers | Additional information |
|---|---|---|---|---|
| Cell line (*Homo sapiens*) | HeLa control/Fsp27-EGFP | other | | Cell line stably expressing Fsp27-EGFP from tet-inducible promoter. Obtained from Koini Lim (David Savage lab). |
| Antibody | Mouse anti-cytochome *c* (monoclonal) | BD Pharmingen | BD Pharmingen: 556432 | (1:250) |
| Antibody | Rabbit anti-TOM20 (polyclonal) | Santa Cruz Biotechnology | Santa Cruz:sc-11415 | (1:250) |
| Antibody | Goat anti-rabbit Alexa Fluor 405 nm | Invitrogen | Invitrogen:A31556 | (1:200) |
| Antibody | Donkey anti-mouse Alexa Fluor 647 nm | Invitrogen | Invitrogen:A31571 | (1:200) |
| Recombinant DNA reagent | hBax-C3-EGFP | Addgene | Addgene:19741 | |
| Chemical compound, drug | ABT-737 | Cayman | Cayman:11501 | assay concentration = 10 µM |
| Chemical compound, drug | Doxycycline | Takara | Takara:631311 | assay concentration = 1 µg/mL |
| Chemical compound, drug | Hygromycin B | Invitrogen | Invitrogen:10687010 | assay concentration = 0.2 µg/mL |
| Chemical compound, drug | Oleic acid | Sigma | Sigma:O3008 | assay concentration = 0.4 mM |
| Chemical compound, drug | Q-VD-Oph | APExBIO | APExBIO:A1901 | assay concentration = 10 µM |
| Commercial assay or kit | Lowicryl HM20 | Polysciences, Inc. | Polysciences, Inc.:15924–1 | |
| Software, algorithm | Correlative Microscopy, MATLAB-based scripts | PMID:22863005; PMID:24275379; Mathworks | | https://www.embl.de/download/briggs/cryoCLEM/index.htm |
| Software, algorithm | IMOD | PMID:8742726 | | http://bio3d.colorado.edu/imod/ |
| Software, algorithm | SerialEM | PMID:16182563 | | http://bio3d.colorado.edu/SerialEM/ |
| Software, algorithm | Ridge Detection 1.4.0 | | | https://imagej.net/Ridge_Detection |
| Other | Specimen carrier, aluminum, B | Engineering Office M. Wohlwend | Engineering Office M. Wohlwend: Art. 1314 | |
| Other | Specimen carrier, copper gold-coated | Engineering Office M. Wohlwend | Engineering Office M. Wohlwend: Art. 1322 | |
| Other | EM grids, copper, 200 mesh carbon support | Agar Scientific Ltd. | Agar Scientific Ltd:S160 | |
| Other | EM grids, copper, 200 mesh, R 3.5/1 holey carbon | Quantifoil | | |
| Other | EM grids, gold, 200 mesh, R 2/2 holey carbon | Quantifoil | | |

*Continued on next page*

*Continued*

| Reagent type (species) or resource | Designation | Source or reference | Identifiers | Additional information |
|---|---|---|---|---|
| Other | MitoTracker Deep Red | Thermo | Thermo:22426 | assay concentration = 20 nM |
| Other | Sapphire disks, 3 mm | Engineering Office M. Wohlwend | Engineering Office M. Wohlwend: Art. 405 | |
| Other | TetraSpeck microspheres, 50 nm | Invitrogen | | custom order, (diluted 1:100 in PBS) |
| Other | TetraSpeck microspheres, 100 nm | Invitrogen | T7279 | (diluted 1:50 in PBS) |
| Other | X-tremeGENE 9 | Roche | Roche:06365787001 | 3 μL reagent: 1 μg DNA |

## Cell culture

HeLa cells for all Bax experiments were grown at 37˚C, 5% $CO_2$ in DMEM, high glucose, GlutaMAX, pyruvate (Thermo 31996) medium supplemented with 10% heat-inactivated FBS (Gibco 10270), 10 mM HEPES, and 1 × NEAA (Thermo 11140). Control HeLa cells for matrix density measurements were grown at 37˚C, 5% $CO_2$ in DMEM, high glucose, GlutaMAX, pyruvate (Thermo 31996) medium supplemented with 10% heat-inactivated, Tet-approved FBS, (Pan Biotech p30-3602), 0.2 μg/mL hygromycin B (Invitrogen 10687010), 10 mM HEPES, and 1 × NEAA (Thermo 11140). Bax/Bak DKO HCT116 GFP-Bax cells were grown at 37˚C, 5% $CO_2$ in McCoy's 5A, GlutaMAX medium (Thermo 36600) supplemented with 10% heat-inactivated FBS (Gibco 10270), 10 mM HEPES, and 1 × NEAA (Thermo 11140). Cell lines were regularly tested for mycoplasma infection using the MycoAlert mycoplasma detection kit (Lonza). Cell lines were authenticated as HeLa and HCT116, respectively, by PCR-single-locus-technology (Eurofins).

## Constructs and reagents

hBax-C3-EGFP (Addgene plasmid 19741) (*Nechushtan et al., 1999*) was used for transient GFP-Bax expression in HeLa cells. Transient transfection was performed using X-tremeGENE 9 (Roche 06365787001) at a ratio of 3 μL of transfection reagent to 1 μg DNA. MitoTracker Deep Red (Thermo 22426) was used for cellular staining at 20 nM. Drug treatments used were ABT-737 at 10 μM (Cayman 11501) and Q-VD-OPh at 10 μM (APExBIO A1901).

## Live-cell confocal microscopy

HeLa cells were grown in two-well chamber slides (iBidi 80286 or Lab-Tek 155380), stained with MitoTracker Deep Red, transfected with 1000 ng hBax-C3-EGFP plasmid and incubated with Q-VD-OPh. Bax/Bak DKO HCT116 GFP-Bax cells were grown in two-well chamber slides (iBidi 80286), stained with MitoTracker Deep Red, and incubated with ABT-737 and Q-VD-OPh. Both cell lines were imaged every 30 min. Imaging was performed with either a Zeiss LSM 710 or a Zeiss LSM880 Spectral confocal microscope each equipped with a 63× Plan Apo oil-immersion objective with NA=1.4. GFP-Bax and MitoTracker Deep Red were excited using 488 and 633 nm lasers, respectively. For both cell lines, the live-imaging experiments were repeated at least three times. Fluorescence images of live cells shown in all figures have been processed with the Smooth function in ImageJ and have been adjusted for contrast individually.

## Immunofluorescence microscopy

HeLa cells were plated onto 13 mm cover glasses (Assistant, 41001113) in a 24-well plate, transfected with 300 ng hBax-C3-EGFP plasmid and incubated with Q-VD-OPh for 16 h. Cells were then fixed with 4% paraformaldehyde in phosphate buffer saline (PBS), pH 7.2 for 30 min. The cover glasses containing cells were removed from the plate, blocked for 1 h in 10% goat serum (Sigma G6767) and 1% Saponin (Sigma 8047-15-2) and incubated overnight at 4˚C with 1:250 mouse monoclonal anti-cytochrome c antibody (BD Pharmingen 556432) and 1:250 rabbit polyclonal anti-TOM20 antibody (Santa Cruz Biotechnology; sc-11415). The samples were then incubated with 1:200 goat

anti-rabbit Alexa Fluor 405 and donkey anti-mouse Alexa Fluor 647 antibodies (Invitrogen; A31556 and A31571, respectively) for 1 hr at room temperature and mounted with ProLong Diamond Anti-fade Mountant (Invitrogen P36965) on an imaging slide. Imaging was performed on a Zeiss LSM 710 confocal microscope with a 63× Plan Apo oil-immersion objective with NA=1.4. Alexa Fluor 405, GFP-Bax, and Alexa Fluor 647 were excited at 405, 488, and 633 nm, respectively. Immunofluorescence images shown in *Figure 1—figure supplement 1* have been adjusted for contrast individually.

## Correlative FM and ET of resin-embedded cells

Correlative microscopy of resin-embedded cells was performed as described in (*Ader and Kukulski, 2017*). In brief, cells were grown on 3 mm sapphire disks (Engineering Office M. Wohlwend, Switzerland) in six-well plates for 24 hr, transfected with 2000 ng hBax-C3-EGFP plasmid and incubated with Q-VD-OPh for 16 hr, stained with MitoTracker Deep Red, and high-pressure frozen using a HPM100 (Leica Microsystems), screened for quality of cell distribution and for GFP-Bax expression by cryo-FM (Leica EM Cryo CLEM, Leica Microsystems) equipped with an Orca Flash 4.0 V2 sCMOS camera (Hamamatsu Photonics) and a HCX PL APO 50× cryo-objective with NA = 0.9. For screening, a 2 × 2 mm montage was taken of green (L5 filter, 250 ms), far red (Y5 filter, 100 ms), and brightfield (50 ms) channels (all filters: Leica Microsystems). Z-stacks (0.5 µm intervals) were collected on regions of interest (0.6 x 0.6 mm) with the same exposure settings. Cells were then freeze-substituted with 0.008% uranyl acetate in acetone and embedded in Lowicryl HM20 using a AFS2 (Leica Microsystems). Resin-embedded cells were sectioned 300 nm thin using a Ultracut E Microtome (Reichert) and a diamond knife (Diatome), and collected on 200 mesh copper grids with carbon support (S160, Agar Scientific Ltd.). As fiducial markers for correlation, 50 nm TetraSpeck microspheres (custom order, Invitrogen) diluted 1:100 in PBS pH 8.4 were adsorbed for 5-10 min to the sections. Fluorescence images were acquired using a TE2000-E widefield fluorescence microscope (Nikon) with a 100× oil-immersion TIRF objective with NA = 1.49. Filters: 89006 ET CFP/YFP/mCherry (Chroma), excitation 560/20, dichroic 89008bs, emission 535/30 for YFP-Parkin and 49006 ET CY5 (Chroma), excitation 520/60, dichroic T660lpxr, emission 700/75 for MitoTracker Deep Red. Fiducial markers were visible in both channels. EM was performed on a Tecnai F20 (FEI) operated at 200 kV. Transmission EM (TEM) images at approximately 100 µm defocus were collected using the montaging function in SerialEM (*Mastronarde, 2005*) at a region of interest, at a pixel size of 1.1 nm. Correlation between fluorescence images and TEM montaged images was performed using the fiducial marker positions as previously described (*Ader and Kukulski, 2017*; *Kukulski et al., 2011*). ET was done in Scanning TEM mode on an axial bright field detector. Tilt series were collected using a high-tilt tomography holder (Fischione Instruments; Model 2020) from approximately -65° to +65° (1° increments) at a pixel size of 1.1, 1.6, 2.9, 3.1, or 4.4 nm. Reconstruction and segmentation were performed using IMOD (*Kremer et al., 1996*). The data set on resin-embedded HeLa cells has been acquired from cells grown on two different sapphire disks, vitrified during the same high-pressure freezing session. The data set on resin-embedded Bax/Bak DKO HCT116 has been acquired from cells grown on one sapphire disk. See also *Supplementary file 1*. Segmentations and figures were made from tomograms acquired at 1.1 nm pixel size. For better visibility in all figures, we filtered tomograms with a nonlinear anisotropic diffusion (NAD) and reduced noise in the shown virtual slices by using a Gaussian filter in IMOD. Fluorescence images of resin sections shown in all figures have been rotated to match orientation of ET virtual slices, and have been adjusted for contrast individually.

## Vitreous sectioning and correlative microscopy of vitreous sections

HeLa cells were grown for 24 hr in six-well plates, transfected with 2000 ng hBax-C3-EGFP plasmid and incubated with Q-VD-OPh for 16 hr, then trypsinized and pelleted. Immediately before trypsinizing, cells were stained with MitoTracker Deep Red. Pellets were maintained at 37 °C while they were mixed 1:1 with 40% Dextran (Sigma) in PBS, pipetted into the 0.2 mm recess of gold-coated copper carriers, covered with the flat side of Aluminum carriers B and high-pressure frozen with a Leica HPM100 (Leica Microsystems). 100 nm-thick vitreous sections were produced at -150°C in a UC6/FC6 cryo-ultramicrotome (Leica Microsystems) using cryotrim 25 and a 35° cryo immuno knives (Diatome). The sections were attached using a Crion antistatic device (Leica Microsystems) to EM grids (R3.5/1, copper, Quantifoil) that were plasma cleaned and had 100 nm TetraSpeck beads (Invitrogen)

diluted 1:50 in PBS adhered to them. To identify areas in the sections that contained GFP-Bax signals and were suitable for cryo-ET, we used the procedure described in (*Bharat et al., 2018*). In brief, grids with vitreous sections were imaged by cryo-FM on the Leica EM Cryo CLEM (Leica Microsystems), equipped with an Orca Flash 4.0 V2 sCMOS camera (Hamamatsu Photonics) and a HCX PL APO 50× cryo-objective with NA = 0.9, in a humidity-controlled room (humidity below 25%). For screening, a 1.5 × 1.5 mm montage was taken of green (L5 filter, 1 s), far red (Y5 filter, 1 s), and brightfield (50 ms) channels (all filters: Leica Microsystems). TetraSpecks were visible in both green and far red. Z-stacks were collected of grid squares of interest (0.3 µm intervals), acquiring green (L5 filter, 3 s), far red (Y5 filter, 3 s), and brightfield (50 ms) channels. Localization of GFP signals in cryo-EM intermediate magnification maps was done by visual correlation, as described in (*Bharat et al., 2018*). Subsequent precise correlation was done using custom MATLAB-based scripts (*Kukulski et al., 2012*; *Schorb and Briggs, 2014*). However, because in many areas of the grids, TetraSpeck fiducial markers were sparse, we instead used the centers of carbon film holes as landmarks for correlation between cryo-FM and cryo-EM images. Fluorescence images of vitreous sections shown in Figures have been rotated to match orientation of cryo-EM images and have been adjusted for contrast individually.

## Cryo-FIB milling

HeLa cells were grown for 24 hr on 200 mesh gold grids with a holey carbon film R2/2 (Quantifoil) in six-well plates and transfected with 2000 ng hBax-C3-EGFP plasmid in presence of Q-VD-OPh. Sixteen hr after transfection, cells were stained with MitoTracker Deep Red, grids were manually backside blotted using Whatman filter paper No. 1 and vitrified using a manual plunger. Bax/Bak DKO HCT116 GFP-Bax cells were grown on grids for 36 hr, stained with MitoTracker Deep Red, and incubated with ABT-737 and Q-VD-OPh for 3 hr before plunge-freezing. Control HeLa cell for measurements of matrix density contained a doxycycline-inducible Fsp27-EGFP construct and were prepared for an unrelated project by treating with 0.4 mM oleic acid (Sigma, O3008) and 1 µg/mL doxycycline (Takara, 631311) for 15 hr before incubation with 1× LipidTox Deep Red (Thermo, H34477) for 1 hr, and then plunge-frozen as described above. Grids were screened for cells with GFP-Bax expression using cryo-FM (Leica EM Cryo CLEM, Leica Microsystems), equipped with an Orca Flash 4.0 V2 sCMOS camera (Hamamatsu Photonics) and a HCX PL APO 50 × cryo-objective with NA = 0.9, in a humidity-controlled room (humidity below 25%). For screening, a 1.5 × 1.5 mm montage was taken of green (L5 filter, 250 ms), far red (Y5 filter, 100 ms), and brightfield (50 ms) channels (all filters: Leica Microsystems). Z-stacks were collected of grid squares of interest (0.5 µm intervals) over the cell volume using the same exposure settings. Cells were cryo-FIB milled to prepare lamellae using a Scios DualBeam FIB/SEM (FEI) equipped with a Quorum cryo-stage (PP3010T), following the protocol described in (*Schaffer et al., 2015*). In brief, grids were coated with an organic Pt compound using the gas injection system for either 8 s at 12 mm working distance or 30 s at 13 mm working distance from a stage tilt of 25°. The stage was then tilted so that the grid was at a 10° angle toward the ion beam for all subsequent steps. The electron beam was used at 13 pA and 5-10 kV to locate cells, 2 kV for subsequent imaging. The ion beam was used at 30 kV and 10 pA for imaging. Rough milling was performed at 30 kV ion beam voltage and 0.5 nA current until a lamella thickness of 5 µm was reached. Subsequently, the current was reduced to 0.3 nA until 3 µm lamella thickness, and further to 0.1 nA until 1 µm lamella thickness. Fine milling to a final lamella thickness of approximately 200 nm was performed either at 30 kV and 30 pA, or 16 kV and 11 pA ion beam setting. The temperature of the cryo-stage was kept at -170 to -180°C and that of anti-contaminators below -190°C.

## Electron cryo-tomography of vitreous sections and cryo-FIB milled lamellae

Montaged images of the entire grid were acquired at low magnification at pixel size of 182.3 nm for vitreous sections and either 190.9 or 99.4 nm for lamella. Intermediate magnification maps of grid squares with vitreous sections or lamellae of interest were acquired at pixel size 5.5 nm. Electron cryo-tomographic tilt-series were collected on a Titan Krios (FEI) operated at 300 kV using a Quantum energy filter (slit width 20 eV) and a K2 direct electron detector (Gatan) in counting mode at a pixel size of 3.7 Å and at a dose rate of ~ 2-4 e⁻/pixel/second on the detector, dependent on sample

thickness. Tilt-series were acquired between ±60° starting from 0° with 1° increment using SerialEM (*Mastronarde, 2005*) following a grouped dose-symmetric acquisition with a group size of four (*Bharat et al., 2018*; *Hagen et al., 2017*), and at 5 µm defocus. A dose of approximately 1.0 to 1.2 e⁻/Å$^2$ was applied per image of the tilt-series. Reconstruction and segmentation were performed using IMOD (*Kremer et al., 1996*). The vitreous section data were acquired on sections produced from one high-pressure frozen pellet of one HeLa cell culture. The HeLa (GFP-Bax overexpression) lamella data were acquired on five different lamellae (each lamella corresponding to one cell) produced from three separate plunge-freezing sessions, thus three separate cell culture experiments. The HeLa control lamella data were acquired on two lamellae from two separate plunge-freezing sessions. The Bax/Bak DKO HCT116 lamella data were acquired on one lamella corresponding to one cell. See also *Supplementary file 1*.

Segmentations shown in *Figure 3M and P* only represent those parts of the membranes that were well visible in the electron cryo-tomograms. Due to the anisotropic resolution of electron tomograms, membranes that are oriented at shallow angles or parallel relative to the section plane are difficult to see. We therefore did not segment regions of mitochondria in which we could not unambiguously determine membrane position or connectivity. Ends of the segmentation are indicated in white in *Figure 3M*. For better visibility in all figures, we used tomograms reconstructed by simultaneous iterative reconstruction technique (SIRT) (10 iterations), binned to a pixel size of 7.5 Å, and reduced noise in the virtual slices shown by using both a 3D Median and a Gaussian filter in IMOD. For Videos, the tomographic volumes were filtered as a whole in IMOD.

## Quantifications and statistical analysis

We estimated the size of outer membrane ruptures (*Figure 4M*) by measuring the shortest distance between the two edges visible in a single virtual slice of the electron tomogram using IMOD. In some cases, parts of the membrane were oriented at oblique angles relative to the tomographic image plane. Due to the anisotropic resolution of electron tomograms, these membrane parts were difficult to discern and therefore rupture sizes could not in all cases be estimated. Rupture size distributions of the three inner membrane morphologies were compared using an ordinary one-way ANOVA with Tukey's multiple comparisons test, assuming that the data are normally distributed (significance shown in *Figure 4M*). The mitochondria diameters used to calculate total surface area were estimated by measuring the furthest distance between outer membranes in a single virtual slice of the electron tomogram using IMOD. Surface area of the whole mitochondrion was calculated with the formula for surface area of a sphere, while surface area of the exposed inner membrane was calculated with the formula for surface area of a spherical cap using the rupture sizes (*Figure 4N*).

For detection of line segments in cryo-ET data of vitreous sections, we selected square areas of 151 nm width from areas correlated to the presence GFP-Bax signal and areas in the cytosol without GFP-Bax signal, found in the same tomogram (*Figure 3—figure supplement 1*). We then selected 100 virtual slices (75.3 nm) from the tomogram that corresponded to cellular volume within the vitreous section. To reduce background noise, we generated stacks of maximum intensity projection images, using 10 consecutive virtual slices per image. These images were binned in the x- and y-dimensions to a pixel size of 1.51 nm to further reduce background noise. The ImageJ Ridge Detection plugin (*Steger, 1998*; *Wagner and Hiner, 2017*) was then run on the stack, using the same parameters (line width, 3.5; high contrast, 230; low contrast, 86; sigma, 1.53; lower threshold, 1.50; upper threshold, 3.00; minimum line length, 16.50 pixels; maximum line length, 35.00 pixels; darkline) for each stack. The total numbers of detected segments for five areas correlated to GFP-Bax were compared to areas without GFP-Bax from the same tomogram using a ratio paired t test, assuming normal distribution (significance shown in *Figure 3—figure supplement 1*).

For estimations of mitochondrial matrix density, we used cryo-ET data of cryo-FIB milled, non-apoptotic HeLa cells acquired as a side product in the context of an unrelated project, which here served as the control. Density ratios were quantified as follows: (1) from each of four tomograms per condition, four virtual slices were selected (16 slices in total for each condition; control and Bax-overexpressing), (2) using ImageJ, the mean pixel gray value was taken from 12 randomly selected areas of 30 nm radius, both within regions of matrix as well as cytosol, in each of the 32 virtual slices, (3) the mean pixel gray values for all 12 areas were then averaged together to get an overall mean value for both matrix and cytosol for each virtual slice, and (4) the ratio of mean matrix pixel gray value to cytosolic gray value was then calculated for each virtual slice (*Figure 5C*). Gray values are assigned

by ImageJ using standard grayscale numerical representation (i.e. a low value is attributed to a darker pixel, and a high value is attributed to a lighter pixel.) These values were compared using a two-tailed, unpaired t test with Welch's correction, assuming that the data are distributed normally (significance is shown in *Figure 5C*).

To measure the angle between ATP synthase heads, contours of three points were made in IMOD at the center of the heads of two neighboring ATP synthases and in the middle of the inner membrane between the two heads. The distances between all three points was measured, and the law of cosines was used to calculate the angle between heads.

To identify ATP synthases in compartments without outer membrane in HCT116 cells, we used IMOD to measure the distances between putative ATP synthase head and membrane in either an intact mitochondrion or the unknown compartment (*Figure 6J*). These values were compared using a two-tailed, unpaired t test with Welch's correction, assuming that the data are distributed normally (significance is shown in *Figure 6J*).

All statistical tests were performed using GraphPad Prism.

# Acknowledgements

We thank Julia Mahamid for instruction and advice on cryo-FIB milling, Giuseppe Cannone, Shaoxia Chen and Christos Savva for EM support, Christopher Russo for advice and help with the Scios microscope, Koini Lim for the HeLa cell line used for acquisition of control tomograms. Work in the group of WK is supported by the Medical Research Council (MC_UP_1201/8). Work in the group of RJY is supported by the NIH National Institute of Neurological Disorders and Stroke Intramural Research Program. NRA was supported by a Marshall Scholarship and the NIH-Oxford-Cambridge Scholars Program.

# Additional information

## Competing interests
Richard J Youle: Reviewing editor, *eLife*. The other authors declare that no competing interests exist.

## Funding

| Funder | Grant reference number | Author |
|---|---|---|
| Medical Research Council | MC_UP_1201/8 | Wanda Kukulski |
| National Institute of Neurological Disorders and Stroke | Intramural Research Progam | Richard J Youle |
| Marshall Scholarship | | Nicholas R Ader |
| National Institutes of Health | Oxford-Cambridge Scholars Program | Nicholas R Ader |

The funders had no role in study design, data collection and interpretation, or the decision to submit the work for publication.

## Author contributions
Nicholas R Ader, Formal analysis, Investigation, Visualization, Methodology, Writing—original draft; Patrick C Hoffmann, Investigation, Methodology, Writing—review and editing; Iva Ganeva, Alicia C Borgeaud, Investigation, Writing—review and editing; Chunxin Wang, Resources, Methodology; Richard J Youle, Conceptualization, Resources, Supervision, Writing—review and editing; Wanda Kukulski, Conceptualization, Formal analysis, Supervision, Investigation, Visualization, Methodology, Writing—original draft

## Author ORCIDs

Nicholas R Ader  http://orcid.org/0000-0001-7744-4484
Patrick C Hoffmann  https://orcid.org/0000-0003-3421-6363
Iva Ganeva  https://orcid.org/0000-0003-3221-2502
Alicia C Borgeaud  https://orcid.org/0000-0002-7231-7424
Chunxin Wang  http://orcid.org/0000-0001-6015-6806
Richard J Youle  https://orcid.org/0000-0001-9117-5241
Wanda Kukulski  https://orcid.org/0000-0002-2778-3936

## Decision letter and Author response

Decision letter https://doi.org/10.7554/eLife.40712.038
Author response https://doi.org/10.7554/eLife.40712.039

## Additional files

### Supplementary files

• Supplementary file 1. Sample sizes from which the analyzed electron tomography data sets were generated. Counts include only samples that have contributed to the final data presented in this study. Additional samples and data have been excluded based on either one or more of the following criteria: poor vitrification/sample quality, poor tilt series acquisition quality, poor tomographic reconstruction, no structure of interest contained in the tomographic volume.
DOI: https://doi.org/10.7554/eLife.40712.023

• Transparent reporting form
DOI: https://doi.org/10.7554/eLife.40712.024

### Data availability

Representative electron tomograms have been deposited at the EMDB (https://www.ebi.ac.uk/pdbe/emdb/index.html/). Entries correspond to: Resin-embedded HeLa cell overexpressing GFP-Bax (EMD-4483), resin-embedded Bax/Bak DKO HCT116 cell expressing GFP-Bax (EMD-4484), vitreous section of HeLa cells overexpressing GFP-Bax (EMD-4486), cryo-FIB milled lamella of HeLa cell overexpressing GFP-Bax (EMD-4490), cryo-FIB milled lamella of control HeLa cell (EMD-4491), cryo-FIB milled lamella of Bax/Bak DKO HCT116 cell expressing GFP-Bax (EMD-4492).

The following datasets were generated:

| Author(s) | Year | Dataset title | Dataset URL | Database and Identifier |
|---|---|---|---|---|
| Ader NR, Hoffmann PC | 2019 | Resin-embedded HeLa cell overexpressing GFP-Bax | https://www.ebi.ac.uk/pdbe/entry/emdb/EMD-4483 | Electron Microscopy Data Bank, EMD-4483 |
| Ader NR, Hoffmann PC, Ganeva I, Borgeaud AC, Wang C, Youle RJ, Kukulski W | 2019 | Resin-embedded Bax/Bak DKO HCT116 cell expressing GFP-Bax | https://www.ebi.ac.uk/pdbe/entry/emdb/EMD-4484 | Electron Microscopy Data Bank, EMD-4484 |
| Ader NR, Hoffmann PC, Ganeva I, Borgeaud AC, Wang C | 2019 | Vitreous section of HeLa cells overexpressing GFP-Bax | https://www.ebi.ac.uk/pdbe/entry/emdb/EMD-4486 | Electron Microscopy Data Bank, EMD-4486 |
| Ader NR, Hoffmann PC, Ganeva I, Borgeaud AC, Wang C, Youle RJ, Kukulski W | 2019 | Cryo-FIB milled lamella of HeLa cell overexpressing GFP-Bax | https://www.ebi.ac.uk/pdbe/entry/emdb/EMD-4490 | Electron Microscopy Data Bank, EMD-4490 |
| Ader NR, Hoffmann PC, Ganeva I, Borgeaud AC, Wang C, Youle RJ, Kukulski W | 2019 | Cryo-FIB milled lamella of control HeLa cell | https://www.ebi.ac.uk/pdbe/entry/emdb/EMD-4491 | Electron Microscopy Data Bank, EMD-4491 |
| Ader NR, Hoff- | 2019 | Cryo-FIB milled lamella of Bax/Bak | https://www.ebi.ac.uk/ | Electron Microscopy |

| mann PC, Ganeva I, Borgeaud AC, Wang C, Youle RJ, Kukulski W | DKO HCT116 cell expressing GFP-Bax | pdbe/entry/emdb/EMD-4492 | Data Bank, EMD-4492 |

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
