## [Decision Letter]

[**Editorial note:** This article has been through an editorial process in which the authors decide how to respond to the issues raised during peer review. The Reviewing Editor's assessment is that all the issues have been addressed.]

Thank you for submitting your article "Molecular and topological reorganizations in mitochondrial architecture interplay during Bax-mediated steps of apoptosis" for consideration by *eLife*. Your article has been reviewed by two peer reviewers, and the evaluation has been overseen by Werner Kühlbrandt as the Reviewing Editor and John Kuriyan as the Senior Editor. The peer reviewers have opted to remain anonymous.

As the Reviewing Editor, I recommend acceptance of the manuscript without major new experimental work. In particular, direct labelling of BAX in cells with a tag that would be visible by electron cryo-tomography strikes me as unrealistic. Orthogonal studies, e.g. comparison to tomography of BAX clusters formed in vitro or mutant studies, may be feasible, if the in vitro system or mutants are established in your laboratory. If this is not the case, I would regard these requests as too onerous. On the other hand, if the experiments can be done within a few months, they would certainly strengthen the manuscript significantly.

In my correspondence with the reviewers I have made the point that the study is by its very nature correlative (hence the name CLEM). Therefore to say that it is correlative (Reviewer 1) cannot be a valid criticism.

The other concerns that require revision and/or responses are included the separate reviews below for your consideration. If you have any questions, please do not hesitate to contact us.

Summary:

The manuscript addresses a central question in molecular cell biology – the role of BAX in the rupture of mitochondria, leading to apoptosis. Dr. Kukulski addresses this question by an experimental tour de force that combines correlative light and electron microscopy, cryo-FIB milling and electron cryo-tomography at the most demanding level.

Both reviewers recognise the potential importance of the work and its interest in particular to the apoptosis field. They provide clear and constructive recommendations to increase the significance of the study and to clarify some points.

Major concerns:

Reviewer 1 asks for additional evidence to support that the BAX "sub-structures" really contain BAX. He/she thinks that, although challenging, cryoEM and cryoET could be combined with immunogold labeling of permeabilised intact cells. Alternatively orthogonal corollary assays, for example their absence in a BAX mutant (e.g L63A) that cannot dimerise, would provide additional support.

Also, EM of a minimal membrane system (e.g. liposomes, GUVs) with recombinant BAX would support that the sub-structures (lines, etc) observed in cells are BAX oligomers. You could also test GFP-BAK- given their functional and structural redundancies, similar sub-structures and changes in mitochondrial morphology would be expected.

Reviewer 2 recommends that you avoid the impression that the interpretation is guided by expectations or by the human tendency to see patterns. In Figure 3 A-H' and Videos 1 and 2, clusters are visualized using CLEM data on vitreous sections; however, to the untrained eye it is not obvious that similar densities are absent from areas lacking a GFP signal, and the red highlights label some of the visible densities, but not all. Based on these data, the authors focus on similar structures in tomograms from FIB-milling, for which there are no fluorescence correlation data (Figure 3I-Q' and Videos 3 and 4) to derive clearer images of putative Bax clusters. To make this analysis more robust and unbiased, the authors should compare proposed Bax clusters with data for regions that appear dense or structured but that do not correspond to GFP signals in CLEM or do not localize adjacent to mitochondria in FIB-milled cells. Optimally, the authors could validate the structural data by an unrelated method, for instance by recapitulating Bax cluster formation in vitro.

Separate reviews (please respond to each point):

*Reviewer #1:*

The current manuscript is an interesting study characterising the BAX apoptotic pore in cells using elegant correlative light/EM. Previous studies have used cryoEM and super resolution imaging to characterise the BAX apoptotic pore, but to my knowledge this is the first to use correlative light and cryoEM in cells undergoing apoptosis. Such studies in cells are particularly challenging and will be of interest to the field. The findings of this study support previous data in model systems that BAX forms large rupture in the mitochondria outer membrane of variable size, and suggest that these ruptures promote swelling of the matrix and deformation of inner membrane cristae. The study also provides evidence that outer membrane rupture can lead to complete expulsion of the inner-membrane-encapsulated matrix analogous to recently described herniations.

My major issue is that the conclusions are largely correlative and based on significant assumptions. Firstly, that the areas in the cytoplasm that were devoid of electron dense ribosomes are BAX aggregates and that the ordered structures within represent BAX oligomeric sub-structures. Secondly, the assumption that the electron dense dimers were ATP synthases. If these conclusions were supported by direct empirical evidence would significantly strengthen the manuscript.

1) Can the authors provide additional evidence to support that the BAX "sub-structures" really contain BAX? These structures will obviously be below the resolution amenable to super-resolution imaging. Although challenging, cryoEM and cryoET could be combined with immunogold labeling of permeabilised intact cells. Alternatively orthogonal corollary assays, for example their absence in a BAX mutant (e.g L63A) that cannot dimerise, would provide additional support.

Also, EM of a minimal membrane system (e.g. liposomes, GUVs) with recombinant BAX would support that the sub-structures (lines, etc) observed in cells are BAX oligomers.

They could also test GFP-BAK- given their functional and structural redundancies, similar sub-structures and changes in mitochondrial morphology would be expected.

2) The very nature of the higher order aggregates on which the EM was performed are late stage, likely significantly downstream of cytochrome c release.

3) Can the authors exclude that the ribosome-exclusion regions are due to proteins released from the inter-membrane space rather BAX aggregates?

4) The authors propose that there is a reduction in density of the mitochondrial matrix following BAX pore formation. They attribute this to matrix swelling, however release of matrix contents could presumably also contribute? Additionally, if the matrix contents become less dense during apoptosis then surely the grey value ratio matrix/cytosol as presented in Figure 5C should decrease in GFP-BAX cells compared to control rather than increase?

Minor Comments:

1) The authors should make it clear in their discussion that a caveat of their study is that the experiments were performed in the presence of capsize inhibitors and so whether the late stage events such as the inner membrane changes occur in cells and in vivo is unclear.

2) The Gillies et al. 2014 study by the Kuwana lab reported cream of BAX albeit ion model membranes and should be cited.

*Reviewer #2:*

Ader et al. analyzed the mechanism of Bax during apoptosis using CLEM and other state of the art microscopic methods. They acquired stunning cryo-ET data of mitochondria in various stages of outer membrane disintegration and inner membrane disorganization. Bax clusters are located adjacent to, and apparently embedded in, outer membrane ruptures. Interestingly, and consistent with recent studies, Bax clusters extend significantly into the cytosol. The authors can identify them in electron micrographs based on correlation data and the exclusion of ribosomes from the area. Using FIB milling to better preserve protein structures, they conclude that Bax forms a three-dimensional network that appears to contain irregular planes and lines. They speculate that Bax clusters could remove outer membrane lipids, thereby progressively removing the outer membrane. The size of outer membrane ruptures correlates with the absence of F1Fo-ATP synthase dimers in adjacent inner membranes and loss of inner membrane curvature. By quantifying the electron density of the matrix, the authors even detect dilution of this innermost mitochondrial compartment, likely due to swelling.

This study makes important contributions to the apoptosis field, in particular with its investigation into the nature of Bax clusters. The structural characterization of Bax clusters is the most exciting new finding, however still somewhat contentious.

Major issues regarding Bax cluster identification:

For the analysis of Bax clusters in electron micrographs, the authors need to avoid the impression that the interpretation is guided by expectations or by the human tendency to see patterns. In Figure 3 A-H' and videos 1 and 2, clusters are visualized using CLEM data on vitreous sections; however, to the untrained eye it is not obvious that similar densities are absent from areas lacking a GFP signal, and the red highlights label some of the visible densities, but not all. Based on these data, the authors focus on similar structures in tomograms from FIB-milling, for which there are no fluorescence correlation data (Figure 3I-Q' and Videos 3 and 4) to derive clearer images of putative Bax clusters. To make this analysis more robust and unbiased, the authors should compare proposed Bax clusters with data for regions that appear dense or structured but that do not correspond to GFP signals in CLEM or do not localize adjacent to mitochondria in FIB-milled cells. Optimally, the authors could validate the structural data by an unrelated method, for instance by recapitulating Bax cluster formation in vitro.

Minor Comments:

1) In the CLEM data shown in Figure 2, GFP-Bax signals are mostly adjacent to ruptured mitochondria, but not localized at the ruptures; in Figure 2E there is a dense region adjacent to a rupture that is marked as a cluster, but Figure 2A shows no GFP fluorescence there. How do the authors explain this?

2) Since the outer membrane ruptures are widened by matrix swelling and inner membrane flattening, as the authors point out, the calculation of "missing surface area" comes across as a bit misleading since it sounds as if up to 50% of the outer membrane was removed. This should be clarified with text changes. Based on a lack of correlation between missing surface area and mitochondrion size, the authors conclude that inner membrane flattening is not the only factor driving rupture size. However, this conclusion relies on the assumption that cristae size and density per mitochondrial volume are constant, yet they are apparently not constant.

3) The authors analyze the distribution of ATP synthase that is correlated with inner membrane curvature and find that outer membrane rupture and matrix dilution progressively result in loss of inner membrane curvature and ATP synthase in the vicinity. However, in Figure 3O,P,Q and Video 4 there are highly curved inner membranes studded with ATP synthase directly underneath a patch that is annotated for both outer and inner membrane ruptures and a Bax cluster. How do the authors explain this?

4) How could Bax clusters and ribosomes become enclosed by membranes, marked as outer membrane, so as to appear as an internal mitochondrial compartment (Figure 2B,F,J; Figure 3I,J,K)? Related to this, the authors observe internalization of ribosomes in the intermembrane space in particular in mitochondria with multiple separate inner membrane / matrix compartments that surprisingly leave lots of space between them (Figure 2H,L), an unusual observation in itself. Could such membrane rearrangements be a result of dysfunctional mitochondrial dynamics? The authors should discuss these findings.

5) The figure reference in subsection “Drug-induced apoptosis has similar effects on both outer and inner membrane to Bax overexpression” sixth sentence, should probably say "Figure 2—figure supplement 2D-L".

---

## [Author Response]

Reviewer #1:[…] My major issue is that the conclusions are largely correlative and based on significant assumptions. Firstly, that the areas in the cytoplasm that were devoid of electron dense ribosomes are BAX aggregates and that the ordered structures within represent BAX oligomeric sub-structures.

We address the reviewer’s comment on identity and composition of the clusters and their substructures below (major points 1 and 3).

Secondly, the assumption that the electron dense dimers were ATP synthases. If these conclusions were supported by direct empirical evidence would significantly strengthen the manuscript.

ATP synthase complexes have been rigorously characterized by electron cryo-tomography (cryo-ET) of isolated mitochondria (Davies et al., 2012; Davies et al., 2011; Dudkina et al., 2010; Hahn et al., 2016; Strauss et al., 2008) and by single-particle electron microscopy (Hahn et al., 2018; Srivastava et al., 2018). Therefore, ATP synthases can be readily identified based on the following unique set of criteria:

1) Localization on inner mitochondrial membrane

2) Head-like densities positioned at a distance of approx. 12 nm from the inner membrane, facing the matrix lumen, corresponding to the F1 part of ATP synthases.

3) These head-like densities are abundant on cristae, where they are arranged in dimers that form rows along cristae. This unique architecture has been described for ATP synthases of many species in the above cited cryo-ET papers, and can also be observed in our data (for instance in Figure 5F, F’, Videos 3 and 4).

To emphasize the previously described characteristics of ATP synthases densities, we have modified the text in the Results to read:

“As described by cryo-ET of purified mitochondria, densities characteristic for ATP synthases are localized at the ridges of cristae, where their distinct dimerization is thought to contribute to cristae structure (Anselmi et al., 2018; Davies et al., 2012; Dudkina et al., 2010; Strauss et al., 2008).”

On the boundary membranes, such head-like densities were also present facing the matrix lumen, albeit not arranged as rows of dimers (Figure 5J). However, the densities were at the same distance from the inner membrane as those on cristae (distance head-membrane on cristae: 12.07 nm, SD 1.00 nm, N=65; on boundary membrane: 12.54 nm, SD 0.79 nm, N=44). Furthermore, the boundary membrane locally displayed higher membrane curvature where these head-like densities located, compatible with residual curvature after potential disruption of curvature-inducing dimers (Anselmi et al., 2018).

In the manuscript, we assessed the distance between membrane and head-like densities facing the lumen of single-membrane compartments found in HCT116 cells (Figure 6J). There is no outer membrane present to confirm that these compartments correspond to inner mitochondrial membranes. However, the distances between heads and membranes were the same as for ATP synthases on cristae and boundary membranes of the clearly identified inner membranes described above. Furthermore, the densities on the single-membrane compartments were located to highly curved membrane regions, reminiscent of cristae. An additional indication is the presence of dense accumulations in the lumen, as visible in nearby mitochondria and previously described as mitochondrial solid-phase calcium stores (Wolf et al., 2017). Thus, we are confident that these densities represent ATP synthases as well. To account for the fact that they are not identified by *bona fide* inner membrane localization, we have modified the according passages in the Results to read:

“To determine whether these compartments consisted of inner mitochondrial membranes, we tested whether the particles could correspond to ATP synthases …”

“We concluded that these particles were likely ATP synthases, …”

1) Can the authors provide additional evidence to support that the BAX "sub-structures" really contain BAX? These structures will obviously be below the resolution amenable to super-resolution imaging. Although challenging, cryoEM and cryoET could be combined with immunogold labeling of permeabilised intact cells. Alternatively orthogonal corollary assays, for example their absence in a BAX mutant (e.g L63A) that cannot dimerise, would provide additional support.Also, EM of a minimal membrane system (e.g. liposomes, GUVs) with recombinant BAX would support that the sub-structures (lines, etc) observed in cells are BAX oligomers.They could also test GFP-BAK- given their functional and structural redundancies, similar sub-structures and changes in mitochondrial morphology would be expected.

Our correlative microscopy data on resin and vitreous sections localizes the fluorescent signals of GFP-Bax to ribosome-exclusion zones in the cytoplasm, providing conclusive evidence for large amounts of Bax present in those zones (see also response to major point 3). In vitreous sections, we observe that these zones contain structural elements that suggest a higher order organization (see also response to major issue 1 of reviewer 2). We refer to these zones as Bax clusters because the GFP-Bax signals they correlate to have been described before as Bax (or Bak) clusters by scanning confocal and super-resolution microscopy (Grosse et al., 2016; Nasu et al., 2016; Nechushtan et al., 2001). Furthermore, the ribosome-exclusion zones ultrastructurally resemble the Bax clusters described by immunogold-electron microscopy (Nechushtan et al., 2001). However, we do not intend to suggest that these clusters consist exclusively of Bax. We also do not claim to identify Bax molecules or oligomers to be the “sub-structures” that make up the higher order organization of the clusters. In fact, we speculate that it is likely that the clusters and hence the sub-structures are formed through interactions of Bax with outer membrane lipids and/or proteins, and potentially other components from the cytosol or intermembrane space.

We fully agree with the reviewer that biochemical reconstitution experiments are worthwhile to further investigate the structure of the clusters. As we also agree with the reviewer that Bax might not be the only component of the clusters (see major point 3), such experiments would first require a characterization of cluster composition. Since in this manuscript we do not claim to identify Bax as the sole component of the clusters, we believe that reconstitution experiments are beyond the scope of this manuscript.

2) The very nature of the higher order aggregates on which the EM was performed are late stage, likely significantly downstream of cytochrome c release.

We agree with the reviewer that the targets of our ET experiments reflect late stages of Bax accumulation. However, in our system, complete release of cytochrome *c* was occurring at late stages of GFP-Bax foci formation (Figure 1—figure supplement 1). Even in cells with larger clusters of GFP-Bax, cytochrome *c* was often localized to mitochondria and not fully released into the cytosol (Figure 1—figure supplement 1C). We therefore believe the late stages of Bax accumulation are functionally relevant for complete cytochrome *c* release, and we are confident that our CLEM and cryo-ET experiments reflect time points around cytochrome c release. To strengthen this rationale, we have now quantified the relation between Bax foci and release of cytochrome *c* in the immunofluorescence images and included these numbers in the Results section:

“We confirmed by immunofluorescence that these stages coincided with the release of cytochrome c from the mitochondria (Figure 1—figure supplement 1). Of 42 cells expressing GFP-Bax, 9 contained diffraction-limited Bax punctae and displayed no or little cytosolic cytochrome *c* release (Figure 1—figure supplement 1B). Thirty-three cells contained larger GFP-Bax foci, of which 17 displayed partial and 16 complete cytochrome *c* release (Figure 1—figure supplement 1C and D, respectively). Consequently, for our further experiments, we chose 16 hr after GFP-Bax transfection as a time point that captures stages around cytochrome *c* release.”

3) Can the authors exclude that the ribosome-exclusion regions are due to proteins released from the inter-membrane space rather BAX aggregates?

As explained in the response to point 1, we indeed think it is likely that in addition to Bax, other proteins and macromolecules are present in the ribosome exclusion zones that we identify by correlative microscopy. We agree with the reviewer that this possibility should be emphasized, and have now added the following to the Discussion:

“By correlative microscopy, we identified GFP-Bax signals to correspond to dense regions of the cytoplasm that we refer to as Bax clusters. While these data suggest that Bax is uniformly distributed within these clusters, it is possible that they contain additional components. These could be molecules originating from the cytosol, the intermembrane space or the outer membrane.”

Nevertheless, as explained in point 1, our high-precision correlative microscopy on resin-embedded cells conclusively demonstrates that GFP-Bax is a constituent of the ribosome-exclusion zones. Firstly, of the 82 GFP-Bax signals we correlated, 77 localized to ribosome-exclusion zones in the cytosol. Secondly, the intensity of GFP-Bax signals appears to correlate with the observed size of the corresponding ribosome exclusion zone (Figure 2). We refer to this observation in the Results section: “More intense GFP-Bax signals corresponded to larger ribosome-exclusion zones (Figure 2F).” Importantly, of the 82 correlated GFP-Bax signals, it was only very weak signals that did not localize to ribosome-exclusion zones. These weak signals could correspond to very small accumulations of Bax that are not large enough to exclude ribosomes, or to fractions of Bax signals that are only partially contained within the section.

Based on these results, we refer to the ribosome-exclusion zones as Bax clusters. We appreciate that the term “Bax cluster” can be interpreted either as a cellular structure which may contain additional components, or as an oligomeric entity solely formed of Bax molecules. Indeed, the terms Bax oligomers and Bax clusters have often been used interchangeably in previous literature. We have therefore amended the following text passage in the Results:

“We thus conclude that these ribosome-exclusion zones in the cytosol comprise the Bax clusters previously observed by immuno-electron, fluorescence and super-resolution microscopy (Grosse et al., 2016; Nasu et al., 2016; Nechushtan et al., 2001; Salvador-Gallego et al., 2016; Zhou and Chang, 2008). We henceforth refer to these cellular structures as Bax clusters.”

4) The authors propose that there is a reduction in density of the mitochondrial matrix following BAX pore formation. They attribute this to matrix swelling, however release of matrix contents could presumably also contribute?

We agree with the reviewer’s suggestion that release of matrix contents could also cause a reduction in density and have amended the Discussion to acknowledge this possibility:

“Matrix dilution could be caused by swelling and dilation of the inner membrane compartment, likely to generate turgor pressure and high membrane tension, which could facilitate cristae unfolding and disruption of the ATP synthase organization. Matrix dilution could also arise from efflux of matrix components in addition, or alternatively, to the volume increase.”

We believe that swelling and dilation of the matrix is likely to be a major reason for the reduction in density, because the low number and shallow protrusion of cristae suggest that cristae in these mitochondria are unfolded. Assuming that the total area of inner membrane in a given mitochondrion does not change, cristae unfolding would have to be coupled to increasing the volume of the compartment.

Additionally, if the matrix contents become less dense during apoptosis then surely the grey value ratio matrix/cytosol as presented in Figure 5C should decrease in GFP-BAX cells compared to control rather than increase?

For our analysis, a low gray value was attributed to a darker pixel, and a high gray value was attributed to a lighter pixel, according to common image format conventions. Therefore, in the case where the matrix is denser (appearing darker in a cryo-ET virtual slice) the pixels would have a lower gray value and the ratio of matrix gray value / cytosol gray value would decrease compared to the control. For clarity, we have re-named the y-axis in Figure 5C to “mean matrix gray value / mean cytosol gray value”.

We have also clarified the corresponding passages in the corresponding Materials and methods section and in the legend of Figure 5C.

“Gray values are assigned by ImageJ using standard grayscale numerical representation (i.e. a low value is attributed to a darker pixel, and a high value is attributed to a lighter pixel.)”

“A low value is attributed to a darker pixel, and a high value is attributed to a lighter pixel.”

Minor Comments:1) The authors should make it clear in their discussion that a caveat of their study is that the experiments were performed in the presence of capsize inhibitors and so whether the late stage events such as the inner membrane changes occur in cells and in vivo is unclear.

We agree with the reviewer that the use of caspase inhibitors is important to note. We have included the following text in the Discussion:

“Furthermore, our experiments were performed in the presence of caspase inhibitors to prevent detachment of cells from the substrate. This strategy allowed to visualize events that might otherwise be very transient, but it may affect timing or extent of some of the events observed.”

2) The Gillies et al. 2014 study by the Kuwana lab reported cream of BAX albeit ion model membranes and should be cited.

We thank the reviewer for pointing out that we missed out this key reference. We have amended the text of the Introduction to include a description of the Bax-induced ruptures observed on purified outer mitochondrial membrane by cryo-EM (Gillies et al., 2015).

“In purified outer mitochondrial membranes, Bax-induced ruptures have been observed by electron cryo-microscopy (cryo-EM) (Gillies et al., 2015).”

Reviewer #2:

[…] This study makes important contributions to the apoptosis field, in particular with its investigation into the nature of Bax clusters. The structural characterization of Bax clusters is the most exciting new finding, however still somewhat contentious.Major issues regarding Bax cluster identification:For the analysis of Bax clusters in electron micrographs, the authors need to avoid the impression that the interpretation is guided by expectations or by the human tendency to see patterns. In Figure 3 A-H' and Videos 1 and 2, clusters are visualized using CLEM data on vitreous sections; however, to the untrained eye it is not obvious that similar densities are absent from areas lacking a GFP signal, and the red highlights label some of the visible densities, but not all. Based on these data, the authors focus on similar structures in tomograms from FIB-milling, for which there are no fluorescence correlation data (Figure 3I-Q' and Videos 3 and 4) to derive clearer images of putative Bax clusters. To make this analysis more robust and unbiased, the authors should compare proposed Bax clusters with data for regions that appear dense or structured but that do not correspond to GFP signals in CLEM or do not localize adjacent to mitochondria in FIB-milled cells. Optimally, the authors could validate the structural data by an unrelated method, for instance by recapitulating Bax cluster formation *in vitro*.

We agree with the reviewer that a more unbiased analysis of the presence of structural elements in Bax clusters is important. We thank the Reviewer for the suggestion to compare areas correlated to GFP-Bax with areas that without GFP-Bax signals in the correlative cryo-microscopy data from vitreous sections. The observation of structural elements in the Bax cluster identified in this data is indeed the critical link to the identification of Bax clusters in cryo-FIB milled cells lacking precise correlation to fluorescence signals.

Therefore, on the cryo-ET data from vitreous sections, we have now performed a quantitative comparison of regions correlating to GFP-Bax signal and regions of cytosol without GFP-Bax signal. This analysis is based on a ridge-detection algorithm, and the results are displayed in a new figure, Figure 3—figure supplement 1. Importantly, this analysis robustly detected more line or ridge-like segments within GFP-Bax clusters compared to regions absent of GFP-Bax signal. We believe these results strengthen our rationale for suggesting that the clusters display a higher-order organization, and for identifying Bax clusters in the cryo-FIB milled cells based on the presence of similar structural elements, in addition to their localization near ruptured mitochondria.

We have added these findings to the Results:

"We tested if the occurrence of these structural elements was specific to Bax clusters. For that, we compared areas that correlated to the presence of GFP-Bax signal to areas without GFP-Bax signal within the same tomogram, using an image analysis tool that detects ridge-like segments (see Materials and methods, and Figure 3—figure supplement 1) (Steger, 1998; Wagner, 2017). We consistently found that the number of detected segments was higher in areas corresponding to GFP-Bax signals than in the areas that did not correlate to GFP-Bax signals (Figure 3—figure supplement 1H).”

We describe how the analysis was performed in the Materials and methods:

“For detection of line segments in cryo-ET data of vitreous sections, we selected square areas of 151 nm width from areas correlated to the presence GFP-Bax signal and areas in the cytosol without GFP-Bax signal, found in the same tomogram. We then selected 100 virtual slices (75.3 nm) from the tomogram that corresponded to cellular volume within the vitreous section. To reduce background noise, we generated stacks of maximum intensity projection images, using 10 consecutive virtual slices per image. These images were binned in the x- and y-dimensions to a pixel size of 1.51 nm to further reduce background noise. The ImageJ Ridge Detection plugin (Steger, 1998; Wagner, 2017) was then run on the stack, using the same parameters (line width, 3.5; high contrast 230; low contrast, 86; σ, 1.53; lower threshold, 1.50; upper threshold, 3.00; minimum line length, 16.50 pixels; maximum line length, 35.00 pixels; darkline) for each stack. The total numbers of detected segments for five areas correlated to GFP-Bax were compared to areas without GFP-Bax from the same tomogram using a ratio paired t test, assuming normal distribution (significance shown in Figure 3—figure supplement 1).”

We have addressed the suggestion to recapitulate Bax clusters *in vitro* in our response to major point 1 of reviewer 1.

Minor Comments:1) In the CLEM data shown in Figure 2, GFP-Bax signals are mostly adjacent to ruptured mitochondria, but not localized at the ruptures; in Figure 2E there is a dense region adjacent to a rupture that is marked as a cluster, but Figure 2A shows no GFP fluorescence there. How do the authors explain this?

There is a GFP-Bax signal at that location in Figure 2A, but it was obscured by the “+” that used to indicate the GFP centroid position. We have increased the size of the fluorescence images and replaced the “+” on the fluorescence image with a circle. We thank the reviewer for pointing out this lack of clarity.

Recent super-resolution studies have reported two types of Bax signals: clusters as well as ring- or arc-like structures (Grosse et al., 2016; Salvador-Gallego et al., 2016). The ring-like structures, suggested to mark the rim of the outer membrane rupture, are reported to be much dimmer than the cytosolic clusters (Grosse et al., 2016). In our CLEM experiments, we detect bright clusters, whereas the ring-like signals are likely below our detection level. Therefore, we do not expect to localize GFP-Bax signals that line the rim of the membrane rupture. The bright clusters have been reported before to localize adjacent to, or on, mitochondria (Grosse et al., 2016; Nasu et al., 2016; Nechushtan et al., 2001) yet not necessarily on the rupture (Grosse et al., 2016).

2) Since the outer membrane ruptures are widened by matrix swelling and inner membrane flattening, as the authors point out, the calculation of "missing surface area" comes across as a bit misleading since it sounds as if up to 50% of the outer membrane was removed. This should be clarified with text changes.

We agree with the reviewer that “missing surface area” can be misleading. We have replaced it by “surface area of the inner membrane that was exposed to the cytosol” or “exposed inner membrane”, in the label of Figure 4N, as well as in the following text passages:

“We could thus estimate the total surface area of these mitochondria, and the surface area of the inner membrane that was exposed to the cytosol due to the rupture. The percentage of mitochondrial surface area occupied by the rupture varied between 2% and 50% (mean total surface area: 1.15 µm^2^, SD 0.41 µm^2^, N = 19; mean surface area of exposed inner membrane: 0.21 µm^2^, SD 0.19 µm^2^, N = 19) (Figure 4N).”

“Surface area of the whole mitochondrion was calculated with the formula for surface area of a sphere, while surface area of the exposed inner membrane was calculated with the formula for surface area of a spherical cap using the rupture sizes (Figure 4N).”

Based on a lack of correlation between missing surface area and mitochondrion size, the authors conclude that inner membrane flattening is not the only factor driving rupture size. However, this conclusion relies on the assumption that cristae size and density per mitochondrial volume are constant, yet they are apparently not constant.

The reviewer is correct in their interpretation that the rupture size will also depend on cristae size and number which can vary among mitochondria, and therefore the variability in rupture size does not necessarily imply that other factors drive rupture size. We have changed the last sentence of the Results section “Inner membrane flattening is most definite at outer membrane ruptures and inner membrane reshaping correlates with rupture size” to avoid this misinterpretation. The concluding sentence now reads:

“Thus, rupture sizes varied largely at a given stage of inner membrane remodeling.”

Furthermore, we have changed the corresponding paragraph in the Discussion to read:

“In this model, the rupture size would depend on the number and surface area of cristae to be unfolded in a given mitochondrion. We indeed observed that rupture sizes varied largely even at a given stage of inner membrane reorganization. This variability in rupture sizes could also be caused by contributions of additional factors potentially impacting on rupture size. One such factor could be the amount of membrane-inserted, accumulated Bax molecules generating tension in the outer membrane (Westphal et al., 2014).”

We have also changed Figure 4N to display the data as the fraction of mitochondrial surface being ruptured. Based on the reviewer’s comment, we think this is a more meaningful measure for the conclusion that rupture sizes vary largely at a given stage of inner membrane rearrangement. The legend for Figure 4N has been updated to reflect this change:

“The percentage of mitochondrial surface area consisting of exposed inner membrane, plotted for mitochondria with unfolded cristae (indicated by schematic in upper right corner). The red lines indicate the mean and the standard deviation.”

3) The authors analyze the distribution of ATP synthase that is correlated with inner membrane curvature and find that outer membrane rupture and matrix dilution progressively result in loss of inner membrane curvature and ATP synthase in the vicinity. However, in Figure 3O,P,Q and Video 4 there are highly curved inner membranes studded with ATP synthase directly underneath a patch that is annotated for both outer and inner membrane ruptures and a Bax cluster. How do the authors explain this?

We thank the reviewer for this observation and comment, as it made us be more precise in our assessment of this point. We had noted that inner membrane segments that are directly exposed by a rupture in the adjacent outer membrane appear smooth, devoid of both cristae and ATP synthases. We now assessed that this is true for 9 of the 12 ruptures identified in the cryo-ET data set, of which 3 are shown in Figure 4. These ruptured mitochondria can contain cristae that are protruding inwards to various extents; some have largely lamellar cristae, while others have short, stub-like cristae (Figure 4M). The mitochondrion shown in Video 4 is classified as one of the first category. The region shown in Figure 3O, P and Q is unique: the inner membrane segment directly underlying the ruptured outer membrane is ruptured as well, rather than being smooth as in the other observed cases. Most of the cristae and ATP synthases are not exposing their intracristae space to the rupture, but have junctions to the boundary membrane in regions that are covered by outer membrane. However, as noted by the Reviewer, the small outer membrane rupture indicated to the left of the double membrane rupture indeed exposes intracristae space to the cytosol. Given that these events are rare, they could reflect rupture widening and inner membrane flattening in progress, and are thus compatible with our model.

To specify our assessment in the Results, we have modified the relevant text passage to read:

“In 11 of the 12 outer membrane ruptures we visualized by cryo-ET, the inner membrane appeared intact with no visible rupture. In only one case, we observed that both outer and inner membrane were ruptured, and a Bax cluster was protruding through the rupture into the mitochondrial matrix (Figure 3O-Q’). In the other 11 cases of ruptured outer membrane, substantial segments of the inner membrane were exposed to the cytosol at the site of the rupture (Figure 4A-C). In nine of these cases, there were no cristae protruding from the exposed inner membrane segment, and no intracristae spaces exposed to the outer membrane ruptures (Figure 4A-C).”

4) How could Bax clusters and ribosomes become enclosed by membranes, marked as outer membrane, so as to appear as an internal mitochondrial compartment (Figure 2B,F,J; Figure 3I,J,K)?

We thank the reviewer for pointing out this observation. Many of the mitochondria display large-scale indentations in their overall shapes. ET is done on 100-300 nm sections through cells, and the images shown in the Figures are virtual slices through the tomographic volume. Depending on the orientation of the section and the image plane relative to the mitochondrial surface indentation, cytosolic content within that indentation may appear enclosed by mitochondrial membrane. Most mitochondria are too large to be contained entirely within a section volume, therefore the overall shape of many mitochondria cannot be determined from a tomogram. However, in most cases, outer and inner membranes can be identified by following their connectivity and appearance throughout the tomographic volume, and thus presumable enclosures can be recognized as cytoplasmic indentations. To clarify these images, we have added Figure 2—figure supplement 1, as well as the following text to the Results section:

“Some of the mitochondria displayed large-scale indentations of their surfaces as they appeared to concave inward. Depending on their orientation within the tomographic volume, these indentations appeared as if cytosolic content was enclosed in a mitochondrion (Figure 2B, Figure 2—figure supplement 1).”

Related to this, the authors observe internalization of ribosomes in the intermembrane space in particular in mitochondria with multiple separate inner membrane / matrix compartments that surprisingly leave lots of space between them (Figure 2H,L), an unusual observation in itself. Could such membrane rearrangements be a result of dysfunctional mitochondrial dynamics? The authors should discuss these findings.

We agree with the reviewer that these observations are interesting findings that we have only described in the Results. We have now added the following text to the Discussion:

“We observed fragmentation of the inner membrane compartments without outer membrane fission. In these instances, the intermembrane spaces were often enlarged and contained ribosomes, indicating influx of cytosolic content into the intermembrane space. Such a mixing of compartment content could potentially play a role in downstream apoptotic events that require interaction between components from both compartments, such as apoptosome formation (Kim et al., 2005; Zhou et al., 2015). In many cases, we observed that cristae locally unfolded into short, tubular protrusions and shallow ridges. These inner membrane remodeling events could be the result of changes in processing and/or activity of OPA1, leading to cristae disassembly and inner membrane fission (Anand et al., 2014).”

5) The figure reference in subsection “Drug-induced apoptosis has similar effects on both outer and inner membrane to Bax overexpression” sixth sentence, should probably say "Figure 2—figure supplement 2D-L".

The reviewer is correct. In the revised manuscript, this becomes Figure 2—figure supplement 2D-L.

Further significant changes to the revised manuscript:

We have modified Figure 1—figure supplement 1 (previously Suppl. Figure S1), in which we have previously used two separate color merges to show the immunofluorescence signal of cytochrome *c*, GFP-Bax and Tom20. We have now used a different color combination to display all three channels in the same merge. We believe that this presentation is clearer and allows a more direct assessment of the relative localization of the signals.

Furthermore, we have deposited representative electron tomograms from each data set discussed in the manuscript to the EMDB. The entries will be released upon publication.